# A Thermodynamic Potential of Seawater in terms of Absolute Salinity, Conservative Temperature and *in situ* Pressure

Trevor J. McDougall[1],  Paul M. Barker[1],  Rainer Feistel[2]  and  Fabien Roquet[3]

[1]School of Mathematics and Statistics, University of New South Wales, Sydney, NSW 2052, Australia

10   [2]Leibniz-Institut für Ostseeforschung, Warnemünde, Germany

[3]Department of Marine Sciences, University of Gothenburg, Sweden

*Correspondence to*: Trevor J. McDougall (Trevor.McDougall@unsw.edu.au)

**Abstract.** A thermodynamic potential is derived for seawater as a function of Conservative Temperature, Absolute Salinity and pressure. From this thermodynamic potential, all the equilibrium thermodynamic properties of seawater can be found, just as all these thermodynamic properties can be found from the TEOS-10 Gibbs function (which is a function of *in situ* temperature, Absolute Salinity and pressure). Present oceanographic practice in the Gibbs SeaWater Oceanographic Toolbox uses a polynomial expression for specific volume (and enthalpy) in terms of Conservative Temperature (as well as of Absolute Salinity and pressure), whereas the relationship between in situ temperature and Conservative Temperature is based on the Gibbs function. This mixed practice introduces (numerically small) inconsistencies and superfluous conversions between variables. The proposed thermodynamic potential of seawater, being expressed as an explicit function of Conservative Temperature, overcomes these small numerical inconsistencies, and in addition, the new approach allows for greater computational efficiency in the evaluation of sea surface temperature from Conservative Temperature. It is also shown that when using Conservative Temperature, the thermodynamic information in enthalpy is independent of that contained in entropy. This contrasts with the cases where either in-situ temperature or potential temperature is used. In these cases, a single thermodynamic potential serves the important purpose of avoiding having to impose a separate consistency requirement between the functional forms of enthalpy and entropy.

# 1 Introduction

## 1.1 Present Practice

The TEOS-10 (the International Thermodynamic Equation of Seawater – 2010, IOC *et al*., 2010) Gibbs function of seawater is a thermodynamic potential whose arguments are Absolute Salinity, *in situ* temperature and pressure. The adoption in 2010 of TEOS-10 as the official description of the thermodynamic properties of seawater came with the recommendation that the observed variables Practical Salinity $S_P$, and *in situ* temperature, together with longitude, latitude and pressure, be used to form Absolute Salinity $S_A$ and Conservative Temperature $\Theta$, and it is these variables, $S_A$ and $\Theta$, that take the place of Practical Salinity $S_P$ and potential temperature $\theta$ in our oceanographic research and in the publication of our results in journals (IOC *et al*., 2010, Valladares et al., 2011a,b, McDougall and Barker, 2011, Pawlowicz *et al*., 2012, Spall *et al*, 2013). Conservative Temperature is proportional to the potential enthalpy of seawater referenced to the pressure of the standard atmosphere (McDougall, 2003, IOC et al., 2010, Graham and McDougall, 2013).

45   The Absolute Salinity variable of TEOS-10 is defined on the Reference-Composition Salinity Scale of Millero *et al.* (2008) as an approximation to the mass fraction of dissolved material in seawater. As described in Pawlowicz (2010, 2011) and Wright *et al.* (2011), while the Gibbs function of a multi-component solution such as seawater should depend on the concentrations of all its constituents, Absolute Salinity on the Reference-Composition Salinity Scale is defined so that its use yields accurate values of the specific volume of seawater.

50   This paper was motivated by the question "is it possible to define a thermodynamic potential in terms of Conservative Temperature rather than, for example, in terms of *in situ* temperature, as is the case for the TEOS-10 Gibbs function of seawater (Feistel, 2008, IAWPS, 2008)?". Progress had already been made towards answering this question in appendix P of the TEOS-10 Manual (IOC *et al.*, 2010) where it was shown that if expressions were available for both the enthalpy and the entropy of seawater as functions of Absolute Salinity, Conservative Temperature, and pressure, then all the thermodynamic properties of

55 seawater could be derived.

   While in situ temperature is an observed variable, its dependence on pressure (even for adiabatic variations of pressure at constant salinity) and its non-conservative nature under turbulent mixing processes, has led to the adoption of Conservative Temperature in order to approximate the "heat content" per unit mass of seawater. It is Conservative Temperature that is now used as the temperature axis of "salinity-temperature" diagrams and as the model's temperature variable in ocean models

60 (McDougall et al., 2021) because it is approximately conserved under mixing processes: the amount of non-conservation is typically two orders of magnitude less than that of potential temperature. In order to facilitate the use of Conservative Temperature in oceanography, Roquet et al. (2015) provided a 75-term polynomial for specific volume, $\hat{v}(S_A, \Theta, P)$, as a function of Absolute Salinity, $S_A$, Conservative Temperature $\Theta$ and pressure $P$, and this polynomial underlies approximately 75 of the 280 algorithms in the Gibbs Seawater (GSW) Oceanographic Toolbox. The hat over a variable indicates that it is

65 being expressed as a function of Conservative Temperature (rather than the in situ absolute temperature $T = T_0 + t$, where $t$ is the in-situ temperature on the Celsius temperature scale and $T_0 = 273.15$K is the Celsius zero point). While the polynomial expression $\hat{v}(S_A, \Theta, P)$ is as accurate in the oceanographic range of salinity as our present knowledge of seawater properties, it does not give exactly the same values for specific volume as are obtained by using the original TEOS-10 Gibbs function. One consequence of this approximation is that there is at present a slight inconsistency in the conversions between different

70 types of temperature variables using the Gibbs function compared with using the Roquet et al. (2015) polynomial $\hat{v}(S_A, \Theta, P)$. For example, the in-situ and potential temperatures, $t$ and $\theta$ respectively (both measured on the Celsius temperature scale), are related through the Gibbs function through the implicit relationship $g_T(S_A, T_0 + t, P) = g_T(S_A, T_0 + \theta, P_r)$ (where the $T$ subscripts denote partial differentiation, $P_r$ is the reference pressure of the potential temperature, and $g(S_A, T_0 + t, P)$ is the

Gibbs function). These temperatures are also related through the forward expression $(T_0 + t)/(T_0 + \theta) = \hat{h}_\Theta(S_A, \Theta, P)/c_p^0$

in terms of the $\Theta$ derivative of enthalpy: note that $\hat{h}_P(S_A, \Theta, P) = \hat{v}(S_A, \Theta, P)$ and that $c_p^0$ and $T_0$ are constants. By "forward expression" we mean that the calculation is performed without doing a series of iterations such as occurs in a Newton-Raphson iterative calculation that is often required in thermodynamic calculations. When the Roquet et al. (2015) polynomial of $\hat{v}(S_A, \Theta, P)$ is used to evaluate $\hat{h}_\Theta(S_A, \Theta, P)$, the differences in temperature are small compared with using the Gibbs function itself, being no larger than $10^{-4}$K, but we would prefer if they did not exist, and the use of the thermodynamic potential of this paper in place of the Gibbs function eliminates both these small inconsistencies as well as the need for superfluous conversions between different temperature variables.

## 1.2 Thermodynamic fundamentals

The First Law of Thermodynamics (see sections 49, 57 and 58 of Landau and Lifshitz (1959) and Appendix B of IOC et al., 2010),

$$\rho\left(\frac{du}{dt} + P\frac{dv}{dt}\right) = -\nabla \cdot \boldsymbol{F}^Q + \rho\epsilon, \tag{1}$$

expresses how the material derivatives of internal energy, $u$, and specific volume, $v$, are related, and how they respond to the local rate of heating by the divergence of the heat flux $\nabla \cdot \boldsymbol{F}^Q$ and by the dissipation of turbulent kinetic energy per unit mass $\epsilon$. In Eq. (1) $t$ stands for time, not in situ temperature; we trust the context makes this use obvious. The symbols used in this paper can be found in Table 1.

Equation (1) illustrates how the work performed by the environment on the fluid parcel due to its change in volume at pressure $P$, -$P$d$v$, changes the internal energy d$u$. The molecular, boundary and radiative fluxes of heat are represented by $\boldsymbol{F}^Q$, and the contribution of the non-conservative nature of Absolute Salinity to the First Law is ignored here; this is discussed in the two paragraphs following Eqn. (A.21.13) in Appendix A.21 of the TEOS-10 Manual, IOC et al. (2010), where this contribution was shown to be a factor of thirty smaller than the $\rho\epsilon$ term in Eq. (1) which itself is routinely ignored. The detailed derivation of the First Law (starting from the conservation of total energy) can be found in Appendix B of IOC et al., 2010.

Clausius (1876) considered the cyclic reversible exchange of heat between a control volume and the environment and inferred that there must be a state variable, which he named entropy, $\eta$, whose total derivative is related to the total derivatives of internal energy d$u$ and volume d$v$ by the following differential relationship,

$$dh - vdP = du + Pdv = Td\eta + \mu dS_A, \tag{2}$$

and the first part of this equation has been added using the definition of specific enthalpy, $h \equiv u + Pv$. This relationship (2) is now called the Fundamental Thermodynamic Relationship (FTR), and the total differentials represent differences between equilibrium states (de Groot and Mazur, 1984, chapter III section 2) that are separated by vanishingly small differences of state variables. This restriction is satisfied for infinitesimally small reversible changes of infinitesimally small seawater parcels,

ensuring that, the in-situ temperature $T$, the relative chemical potential $\mu$ and the pressure $P$ are unambiguously defined. Cullen (1985, section 4.2) explains that Eqns. (1) and (2) apply to "quasi-static" processes that are defined as a series of vanishingly small property changes occurring between a dense succession of "local" equilibrium states. It is only for such "quasi-static" processes that $-Pdv$ can be identified as mechanical work and $Td\eta$ as the heat transfer, for otherwise there are choices to be made of the values of $P$ and $T$, choices that would introduce errors into Eqns. (1) and (2). The infinitesimally

small differences $dh$, $dP$, $d\eta$ and $dS_A$ in Eqn. (2) need not only represent difference in time between successive states but may equally well represent difference between states that are well separated in space and time. The key feature is to realize that, for example, when Absolute Salinity and pressure are both constant, the temperature $T$ is unambiguously defined in all three parts ($h_T$, $T$, and $\eta_T$) of the differential equation $h_T(S_A, T, P) = T\eta_T(S_A, T, P) = c_p(S_A, T, P)$. Bearing in mind this type of restriction, the First Law of Thermodynamics, Eqn. (1), and the FTR, Eqn. (2), may be combined into the following form of

the First Law,

$$\rho\left(\frac{dh}{dt} - v\frac{dP}{dt}\right) = \rho\left(\frac{du}{dt} + P\frac{dv}{dt}\right) = \rho\left(T\frac{d\eta}{dt} + \mu\frac{dS_A}{dt}\right) = -\nabla \cdot \boldsymbol{F}^Q + \rho\epsilon. \tag{3}$$

This version of the First Law may be combined with the equation for the conservation of mass (the so-called "continuity" equation, $\partial\rho/\partial t + \nabla \cdot (\rho\boldsymbol{u}) = 0$ where the bold $\boldsymbol{u}$ is the velocity vector) and rearranged into the form,

$$\rho\frac{d\eta}{dt} = \frac{\partial}{\partial t}(\rho\eta) + \nabla \cdot (\rho\boldsymbol{u}\eta) = -\nabla \cdot \left(\frac{1}{T}\boldsymbol{F}^Q - \frac{\mu}{T}\boldsymbol{F}^S\right) + \boldsymbol{F}^Q \cdot \nabla\left(\frac{1}{T}\right) + \boldsymbol{F}^S \cdot \nabla\left(-\frac{\mu}{T}\right) + \frac{\rho\epsilon}{T}. \tag{4}$$

In doing this rearrangement we have used the evolution equation of Absolute Salinity

$$\rho\frac{dS_A}{dt} = \frac{\partial}{\partial t}(\rho S_A) + \nabla \cdot (\rho\boldsymbol{u}S_A) = -\nabla \cdot \boldsymbol{F}^S, \tag{5}$$

Where $\boldsymbol{F}^S$ is the flux of Absolute Salinity caused by molecular diffusion. This form (5) of an evolution equation for a variable is the "conservative" form, because the right-hand side of this equation is minus the divergence of a molecular flux (see the formal definition of a conservative variable, Eqn. (A.8.1) of the TEOS-10 Manual, IOC et al., 2010). Using Gauss' integral

theorem, it is concluded that the total amount of such a variable in the ocean is then set only by the flux of the variable at the ocean boundaries. Since mixing occurs between fluid parcels only when the fluid parcels are brought together to the same location and therefore same pressure, it follows that apart from the warming due to the dissipation of turbulent kinetic energy, enthalpy is a conservative quantity during an individual mixing process. This is the single most important fact about thermodynamics of importance to physical oceanography. Graham and McDougall (2013) have exploited this fact and have

further particularized it by saying that "apart from the warming caused by the dissipation of turbulent kinetic energy, potential enthalpy referenced to the pressure of the mixing process is conserved during the mixing process". The reasons why we can make this statement are that (i) enthalpy enters Eqn. (3) as density times the material derivative of enthalpy and (ii) mixing processes occur at constant pressure. We are not able to make the corresponding statement about entropy because it enters Eqn. (3) as density times $T$ times the material derivative of entropy (and also because of the non-constancy of $\mu$). The presence

of this multiplicative factor, $T$, is key to explaining why enthalpy is an isobaric-conservative variable while entropy is not.

We note that there is a fundamental difference in the language and symbols used in thermodynamics versus in fluid dynamics. As we have noted, the FTR, Eqn. (2), applies only to reversible processes and yet the FTR has been combined with the First Law of Thermodynamics, Eqn. (1), to arrive at Eqns. (3) and (4) which are written in typical fluid dynamics form using material derivatives. There is a disconnect here; a disconnect that is common in the literature and is the source of much confusion. In fluid dynamics we do not require mixing processes to occur only for an instant and then to have these process switch off while the fluid slowly comes to thermodynamic equilibrium (as would be required to technically obey the thermodynamic restrictions associated with the FTR which we have used). Rather, in fluid dynamics we imagine the mixing processes and the dissipation of turbulent kinetic energy to occur continuously. Moreover, a state of thermodynamic equilibrium has spatially uniform fields of in situ temperature and chemical potential, and such a state is not what we observe or expect in the ocean which is mixed by turbulent mixing processes (see the discussion of this point on the last page Appendix B of IOC et al. (2010)). Hence it is clear that the restrictions associated with use of the FTR are not fulfilled when we combine it with the First Law and write the result using fluid dynamic notation and interpretation as though it might apply to the real ocean. We conclude that there are small thermodynamic inconsistencies involved with combining the FTR and the First Law into the forms of Eqns. (3) and (4). This same inconsistency is common to all advanced thermodynamics textbooks and is rarely discussed; a rare mention of the issue appears on the last page of section 49 of Landau and Lifshitz (1959). Importantly, we point out below (in the paragraph that contains our Eqn. (6)) that in physical oceanography we do not need to use the evolution of entropy as it appears in Eqns. (3) and (4), but rather we exploit the fact that entropy is a function only of state variables and so can be expressed in the functional form $\breve{\eta}(S_A, h, P)$. This sidesteps the otherwise annoying conceptual issues that would arise when applying fluid mechanics concepts and fluid mechanical mathematical nomenclature (such as material derivatives) to the FTR where the same symbols have a different and more restrictive meaning.

A test of the conservative nature (or otherwise) of an oceanographic variable is to consider the turbulent mixing of two seawater parcels. If the total amount of the variable in the final mixed product is the sum of the amounts in the two original parcels, then the variable is conservative. This is rigorously true for enthalpy in an isobaric mixing process (apart from the dissipation of turbulent kinetic energy which needs to be budgeted separately) and is close to being true of Conservative Temperature (McDougall, 2003, Graham and McDougall, 2013).

Temporarily setting aside the reservations we have outlined above with the evolution expressions for entropy in the forms Eqns. (3) and (4), it is customary to note from Eqn. (4) that entropy is not a conservative variable because of the three terms $\mathbf{F}^Q \cdot \nabla(1/T)$, $\mathbf{F}^S \cdot \nabla(-\mu/T)$, and $\rho\epsilon/T$. The Second Law of Thermodynamics can be stated in many forms, and when considering the mixing of a pair of fluid parcels, the Second Law requires that the entropy of the final mixture must be not less than the sum of the entropies contained in the initial two fluid parcels. This is clearly true for the last term in Eqn. (4) because the dissipation of turbulent kinetic energy, $\epsilon$, is always non-negative. The non-negative production of entropy means that the terms in Eqn. (4) involving the molecular fluxes of heat $\mathbf{F}^Q$ and salt $\mathbf{F}^S$, namely $\mathbf{F}^Q \cdot \nabla(1/T)$ and $\mathbf{F}^S \cdot \nabla(-\mu/T)$, also need to be non-negative, and this requirement is shown by Landau and Lifshitz (1959) to be satisfied when the Gibbs function, $g$, satisfies $g_{TT} < 0$ and $g_{S_A S_A} > 0$. The TEOS-10 Gibbs function of seawater satisfies this thermodynamic stability condition.

To understand and quantify the non-conservative production of entropy when turbulent mixing occurs between different seawater parcels, a different approach is required because the production terms $\boldsymbol{F}^Q \cdot \nabla(1/T)$ and $\boldsymbol{F}^S \cdot \nabla(-\mu/T)$ in Eqn. (4) involve complicated products of the gradients of in situ temperature, of pressure, and of salinity, bearing in mind that the molecular fluxes of heat and salt contain contributions from baro-diffusion, and the Soret and Dufour effects (see Appendix B of IOC et al., 2010). These products of gradients would need to be averaged over the time and space scales of the turbulent mixing event. Such a formidable averaging task has never been undertaken and is probably impossible. Fortunately, there is a much simpler way of evaluating the non-conservative production of entropy due to turbulent mixing, namely, to exploit the fact that entropy is a state variable, so that it can be expressed as a function of salinity, enthalpy, and pressure, $\breve{\eta}(S_A, h, P)$ (here the cup over a variable's name indicates that it is being expressed as a function of enthalpy, $h$). Graham and McDougall (2013) used this approach to show that the irreversible production of entropy, $\delta\eta$, that occurs when two seawater parcels of equal mass mix to completion is, to lowest expansion order,

$$\delta\eta \ = \ -\tfrac{1}{8}\big\{\breve{\eta}_{hh}(\Delta h)^2 + 2\breve{\eta}_{hS_A}\Delta h\Delta S_A + \breve{\eta}_{S_AS_A}(\Delta S_A)^2\big\}, \tag{6}$$

where $\Delta h$ and $\Delta S_A$ are the differences between the values of enthalpy and Absolute Salinity of the initial seawater parcels. Graham and McDougall (2013) also developed the evolution equation for entropy in the presence of turbulent epineutral and dianeutral turbulent mixing (their Eqn. 48), and this work is summarised in section A.16 of IOC et al. 2010. There it is shown that the sign-definite nature of the production of entropy for the turbulent mixing process places exactly the same requirements on the Gibbs function of seawater as does molecular diffusion, namely that $g_{TT} < 0$ and $g_{S_AS_A} > 0$. Using this method which recognises that entropy is a state variable so that it is possible to express entropy in the functional form $\breve{\eta}(S_A, h, P)$, largely circumvents the theoretical difficulty of using the FTR in real fluid situations that are clearly not in a state of thermodynamic equilibrium.

The $Td\eta$ term in Eqn. (2) describes the exchange of an infinitesimally small amount of heat and constitutes the original definition of entropy by Clausius (1876), so that, for example, if a seawater parcel is heated reversibly at constant pressure and salinity, this input of heat is equal to both $dh$ and $Td\eta$. The last term, $\mu dS_A$, describes the influence of changes in Absolute Salinity on enthalpy at constant entropy and pressure, that is, $\mu$ is the relative chemical potential defined by $\mu = \partial h/\partial S_A|_{\eta,P}$, which is also given by $\mu = \partial u/\partial S_A|_{v,P}$. While the FTR relates the total derivatives of the several thermodynamic quantities only for thermodynamically reversible processes, importantly all of enthalpy, internal energy, specific volume, entropy, and relative chemical potential are state variables so that they can be expressed as functions of, for example, $(S_A, T, P)$. That is, after a series of irreversible processes (such as events in which turbulent kinetic energy is dissipated), the differences in these variables are still given by the differences in their functional expressions. Specifically, knowing the values of salinity, temperature and pressure both before and after the occurrence of an irreversible process, the difference in entropy after this irreversible process is given by the difference between the final and initial values of $\eta(S_A, T, P)$, under the assumption that the fluid sample under consideration is at thermodynamic equilibrium both before and after that process. Given this, the nature of the processes occurring between the initial and final times is irrelevant.

In practice the FTR is used extensively in the construction of the thermodynamic potentials that describes seawater, so that all the thermodynamic variables are related to each other using equations that apply for reversible processes. Because each of these thermodynamic variables are state variables, the use of the FTR is justified; its use essentially finds a route through parameter space caused by a series of reversible processes, even though there are many other ways of traversing between two $(S_A, T, P)$ states, specifically, ways that involve irreversible processes. Thermodynamic state variables, by definition, never depend on the process history that has led to the actual state. Rather, "the actual state of the world depends only on the most recent past, without being directly influenced, so to speak, by the memory of the distant past" wrote Henri Poincaré in a report to the International Congress of Physics in 1900 (Poincaré and Goroff, 1993, pI18).

Two important characteristics of oceanographic variables are (i) whether they are "potential" variables, and (ii) whether they are "conservative" variables: these characteristics are discussed at length in sections A.8 and A.9 of IOC et al., 2010. For example, Absolute Salinity is a potential variable since if the salt flux divergence, $\nabla \cdot \boldsymbol{F}^S$, is zero then the salinity of a fluid parcel is unchanged even though its pressure may vary: this follows from the conservation equation of Absolute Salinity, $\rho \mathrm{d}S_A/\mathrm{d}t = -\nabla \cdot \boldsymbol{F}^S$ of Eqn. (5) (where, again, we are neglecting the influence of the non-conservative source term of Absolute Salinity). A "potential" variable is independent of pressure when the pressure change occurs isentropically and without change in Absolute Salinity. From Eqn. (3), if in addition to being isohaline, both $\nabla \cdot \boldsymbol{F}^Q = 0$ and $\epsilon = 0$ so that there is no net flux of heat across the boundaries of the fluid parcel and no dissipation of turbulent kinetic energy inside the parcel, then entropy $\eta$ is also constant, showing that entropy also has the "potential" property; indeed this is a fundamental definitional property of entropy. Potential enthalpy, potential density, and potential temperature, $\theta$, all have the "potential" property, by construction.

Since Conservative Temperature $\Theta$ is defined as being proportional to potential enthalpy, $h(S_A, T_0 + \theta, P_0)$, it is also a "potential" variable and can be regarded as a function $\widetilde{\Theta}(S_A, \theta)$ of only $S_A$ and $\theta$. It follows that entropy, which is also a "potential" variable, obeys $\eta = \eta(S_A, T_0 + t, P) = \eta(S_A, T_0 + \theta, P_0) = -g_T(S_A, T_0 + \theta, P_0)$ and so can be expressed as a function of $S_A$ and $\Theta$ only, $\eta = \hat{\eta}(S_A, \Theta)$, and is not a separate function of in-situ pressure. Note that molecular diffusion acts primarily to flux heat down the temperature gradient (up the gradient of $1/T$) and, in the presence of a pressure gradient (such as that caused by the gravitational hydrostatic balance) does not act to eliminate entropy gradients. In contrast, turbulent mixing, by exchanging fluid parcels, acts to flux "potential" properties (such as entropy and Conservative Temperature) down the gradients of these "potential" variables, and, in the presence of a pressure gradient, establishes a gradient of in situ temperature.

**1.3 An introduction to thermodynamic potentials**

The Fundamental Thermodynamic Relationship of Eqn. (2) can be regarded as an expression for the total derivative of enthalpy when it is expressed as a function of $(S_A, \eta, P)$ and the three partial derivatives with respect to these variables are $\mu, T$ and $v$. Thermodynamically speaking, this form of enthalpy, namely $\overset{\rightharpoonup}{h}(S_A, \eta, P)$, is the most natural thermodynamic potential of seawater because turbulent mixing events in the ocean occur at constant pressure rather than at constant volume (here the

235 bracket over a variable's name indicates that it is being expressed as a function of entropy). This is because for mixing between seawater parcels to occur, these parcels need to be in contact with each other, irrespective of whether the seawater parcels have previously travelled through physical space vertically or along a surface of constant potential density. This need for identical geolocation is why turbulent mixing between a pair of fluid parcels occurs at a given value of pressure. The "heat-like" argument of $\widetilde{h}(S_A, \eta, P)$, namely entropy, is a "potential" variable, and this "potential" property leads to simple expressions

for quantities such as the adiabatic and isentropic compressibility, $\kappa = -\widetilde{h}_P^{-1}\,\widetilde{h}_{PP}$. But entropy, $\eta$, is neither an observed quantity (c.f. in situ temperature $T$), nor is it an almost conservative quantity (c.f. Conservative Temperature $\Theta$). The Gibbs function $g(S_A, T, P)$ has proven to be a practically more useful thermodynamic potential than $\widetilde{h}(S_A, \eta, P)$ because its "heat-like" argument, $T$, is an observed quantity, even though $T$ is neither a "potential" variable nor is it an almost conservative variable. We note, from the FTR, that an alternative to $\widetilde{h}(S_A, \eta, P)$ as a thermodynamic potential is internal energy as a

function of $(S_A, \eta, v)$ where specific volume (or density) takes the place of pressure as an independent variable and the partial derivatives are $\mu, T$ and $-P$. For completeness it may be mentioned that the thermodynamic potential of pure water, which is part of TEOS-10, is a Helmholtz function expressed as a function of $(T, v)$ which permits the joint description of liquid and gaseous water by a single mathematical expression (Wagner and Pruß, 2002).

  Importantly, all thermodynamic potentials obey the three general criteria which characterise axiomatic systems (Feistel,

2008, 2018). That is, thermodynamic potentials must exhibit *consistency* (that is, they exclude the possibility of deducing two different mathematical expressions for the same property), *independence* (that is, they prevent any derived function from being deducible from another one) and *completeness* (that is, they provide an equation for every equilibrium thermodynamic bulk property). For an arbitrary given thermodynamic property equation, the validity of these criteria is not trivially fulfilled and needs to be demonstrated in order to regard that equation a thermodynamic potential. The new thermodynamic potentials of

this paper do obey these three essential criteria.

## 1.4 A guide to this paper

  In this paper we derive a new thermodynamic potential of seawater, $\hat{\phi}(S_A, \Theta, P)$, whose "heat-like" variable is Conservative Temperature, $\Theta$, which, while not being a measured quantity, is a "potential" variable, and is also close to being 100% conservative. We also find a new thermodynamic potential, $\tilde{\psi}(S_A, \theta, P)$, whose "heat-like" variable is potential temperature,

$\theta$. Of the three desirable attributes of the "heat-like" argument of a thermodynamic potential, namely (i) being an observed quantity, (ii) being a "potential" variable, and (iii) being nearly conservative, $\Theta$ has two of these attributes, while all of $T, \theta$ and $\eta$ have only one attribute each.

  In section 2 we compare two ways of defining the properties of seawater. In one way we claim to have knowledge of both enthalpy and entropy as functions of in situ temperature, that is, we claim to know both $h(S_A, T, P)$ and $\eta(S_A, T, P)$, while in

the other case we claim to know enthalpy and entropy as functions of Conservative Temperature, that is, we claim to know both $\hat{h}(S_A, \Theta, P)$ and $\hat{\eta}(S_A, \Theta)$. We show that the $\hat{h}(S_A, \Theta, P)$ and $\hat{\eta}(S_A, \Theta)$ pair provides a clean separation of the heat and

buoyancy information, namely specific volume, internal energy, isentropic compressibility and sound speed, all of which are found from $\hat{h}(S_A, \Theta, P)$ alone, while the information in $\hat{\eta}(S_A, \Theta)$ is needed to relate $\Theta$ to the other temperature variables and to evaluate the chemical potentials. This contrasts with the $(S_A, T, P)$ case where these same thermodynamic properties, namely specific volume, internal energy, isentropic compressibility, and sound speed, all depend on both $h(S_A, T, P)$ and $\eta(S_A, T, P)$. Also, the information in $h(S_A, T, P)$ and $\eta(S_A, T, P)$ is not independent of each other since these functions need to satisfy the constraint $h_T = T\eta_T$. In contrast, there is no such consistency requirement between the $\hat{h}(S_A, \Theta, P)$ and $\hat{\eta}(S_A, \Theta)$ functions because in the equation, $\hat{h}_\Theta(S_A, \Theta, P) = T\,\hat{\eta}_\Theta(S_A, \Theta)$, in-situ temperature $T$ is not an independent variable in this case.

In section 3 we find a new thermodynamic potential $\hat{\phi}(S_A, \Theta, P)$ from which both enthalpy $\hat{h}(S_A, \Theta, P)$ and entropy $\hat{\eta}(S_A, \Theta)$ can be found, and we argue that this thermodynamic potential is as fundamental as the Gibbs function, and that the approach using $\hat{\phi}(S_A, \Theta, P)$ has several advantages over the Gibbs function, owing to the facts that (i) Conservative Temperature is a "potential" variable, and (ii) it is an almost conservative variable. The formation of the combined logarithm and polynomial construction of the expression for $\hat{\eta}(S_A, \Theta)$ is described in section 4, while section 5 describes how the new thermodynamic potential is used in observational oceanography and in numerical ocean models. The paper ends with the conclusions section, section 5.

## 2 Thermodynamic potentials versus knowledge of both enthalpy and entropy

### 2.1 Four known thermodynamic potential functions

The FTR, Eqn. (2), in its original form, $du + Pdv = Td\eta + \mu dS_A$, is an expression for the total derivative of internal energy in terms of the total derivatives of its natural (or conjugate) variables, Absolute Salinity, entropy and specific volume. This describes the thermodynamic potential $u(S_A, \eta, v)$ and we describe this as the most basic, or "original" thermodynamic potential because it follows from the original form of the FTR. Enthalpy is obtained through the Legendre transformation of internal energy by adding the product of pressure and specific volume, $h \equiv u + Pv$, so that, from the FTR we have $dh - vdP = Td\eta + \mu dS_A$, which is equivalent to the total differential of the thermodynamic potential $h(S_A, \eta, P)$, written in terms of its canonical independent variables. Note that the original form of the FTR, namely $du + Pdv = Td\eta + \mu dS_A$, can be deduced from the expression for the total derivative of enthalpy, $dh = Td\eta + \mu dS_A + vdP$, *if and only if* one also knows that enthalpy is defined in terms of internal energy by $h \equiv u - Pv$.

The Gibbs function $g(S_A, T, P)$ is found from the Legendre transformation of enthalpy by subtracting the product of entropy and absolute temperature, $g \equiv h - T\eta \equiv u + Pv - T\eta$. The total differential of the Gibbs function, $dg = dh - \eta dT - Td\eta$, can be found from the FTR (Eqn. 2) to be

$$dg = \mu dS_A - \eta dT + vdP, \tag{7}$$

with the three partial derivatives of $g(S_A, T, P)$ being $\mu$, $-\eta$ and $v$. We can think of the Gibbs function being formed from laboratory-derived measurements of these three partial derivatives. Note that the FTR can only be deduced from this

expression for the total derivative of the Gibbs function *if and only if* one also knows that the Gibbs function is defined ini terms of enthalpy by $g = h - T\eta$. The Helmholtz free energy $f(S_A, T, v)$ is found from the Legendre transformation of internal energy by subtracting the product of entropy and absolute temperature, $f = u - T\eta$. The total differential of the Helmholtz free energy is $df = \mu dS_A - \eta dT - Pdv$. Again, the FTR can only be deduced from this expression for the total derivative of the Helmholtz free energy *if and only if* one also knows that $f = u - T\eta$. The Gibbs function and the Helmholtz free energy are the thermodynamic potentials that prove useful for describing phase transitions because they both have in-situ temperature as an independent variable, and in-situ temperature is common to both phases during an equilibrium phase transition.

Each of the thermodynamic potentials $h(S_A, \eta, P)$, $g(S_A, T, P)$, and $f(S_A, T, v)$ follow from the original thermodynamic potential $u(S_A, \eta, v)$ by a Legendre transformation (and in the case of the Gibbs function, via a sequence of two such transformations) which have the effect of changing the natural (or canonical) independent variables of each thermodynamic potential. In each of these three cases, the original form of the FTR is not deducible from the differential expression of the new thermodynamic potential unless one also knows how the thermodynamic potential is defined in terms of internal energy. In the present paper we present a new thermodynamic potential of seawater, and even though its derivation does not rely on a Legendre transformation, it has the same important characteristic described above, namely that if one knows both the definition of the new thermodynamic potential and the expression for its total differential, then the FTR follows, with the detailed proof of this to be found in Appendix D.

## 2.2 The case of $h(S_A, T, P)$ and $\eta(S_A, T, P)$

The discussion of the derivation, definition and use of the Gibbs function can be approached via a slightly different line of reasoning. We introduce this alternative line of reasoning because it resonates with the same line of reasoning that we use to derive/justify the thermodynamic potential $\hat{\phi}(S_A, \Theta, P)$ of this paper. In this alternative way of approaching the Gibbs function, one again takes $\mu(S_A, T, P)$, $\eta(S_A, T, P)$ and $v(S_A, T, P)$ to be known functions of seawater, but instead of forming a Gibbs function $g(S_A, T, P)$ according to its total differential, Eqn. (7), we instead form the total derivative of enthalpy in the functional form $(S_A, T, P)$, by substituting the total differential of entropy, $d\eta = \eta_{S_A} dS_A + \eta_T dT + \eta_P dP$, into the FTR, obtaining,

$$dh = \left(\mu + T\eta_{S_A}\right)dS_A + T\eta_T dT + (v + T\eta_P)dP, \tag{8}$$

with the three partial derivatives of $h(S_A, T, P)$ being $\left(\mu + T\eta_{S_A}\right) = (\mu - T\mu_T)$, $T\eta_T$ and $(v + T\eta_P) = (v - Tv_T)$ respectively. We can think of enthalpy being formed from these three partial derivatives using laboratory-derived measurements of $\mu(S_A, T, P)$, $\eta(S_A, T, P)$ and $v(S_A, T, P)$. Note that the FTR in the original form, $du + Pdv = Td\eta + \mu dS_A$, follows from this expression for the total derivative of enthalpy by using the total differential of entropy, $d\eta = \eta_{S_A} dS_A + \eta_T dT + \eta_P dP$, as well as the knowledge of the definition of enthalpy in terms of internal energy, $h \equiv u + Pv$. Having formed $h(S_A, T, P)$ by integrating its differential definition, Eqn. (8), and also separately knowing $\eta(S_A, T, P)$, all the thermodynamic

properties can be found. Despite that, however, the combination of $h(S_A, T, P)$ and $\eta(S_A, T, P)$ is not fully equivalent to a thermodynamic potential as this function pair violates the criterion of *independence*. This is evident from the heat capacity for which two different equations can be found,

$$c_P = \left(\frac{\partial h}{\partial T}\right)_{S_A, P} = T\left(\frac{\partial \eta}{\partial T}\right)_{S_A, P}. \tag{9}$$

Therefore, any suitable thermodynamic potential must intrinsically ensure the validity of the consistency condition between
335 enthalpy and entropy,

$$\left(\frac{\partial h}{\partial T}\right)_{S_A, P} \equiv T\left(\frac{\partial \eta}{\partial T}\right)_{S_A, P}. \tag{10}$$

This identity holds for the TEOS-10 Gibbs function, as can be deduced from Eq. (8) which relies on the differential form, Eq. (7), of the Gibbs function, and its definition in terms of enthalpy and entropy, $g = h - T\eta$.

The last step in this alternative narrative that leads to the Gibbs function is to note that it is more convenient to combine
the knowledge contained in $h(S_A, T, P)$ and $\eta(S_A, T, P)$ into the single function, $g = h - T\eta$, whose $T$ derivative gives $-\eta$ (using $h_T = T\eta_T$), and enthalpy can then be found by simply adding $T\eta$ to $g = h - T\eta$.

Comparing the traditional with the alternative reasoning surrounding the Gibbs function, we see that via the traditional approach, in order to deduce at the FTR from knowledge of the Gibbs function one needs to know both (i) how the Gibbs function is found from the observed data, namely, the differential expression Eqn. (7), as well as (ii) the definition of the Gibbs
function in terms of enthalpy and entropy, $g = h - T\eta$. Similarly, with the alternative approach of arriving at the Gibbs function, use of the same observed data of of $\mu(S_A, T, P)$, $\eta(S_A, T, P)$ and $v(S_A, T, P)$ to define specific enthalpy according to Eqn. (8), also needs knowledge of how enthalpy is related to internal energy ($h \equiv u + Pv$) in order to arrive at the FTR. In this alternative approach both entropy $\eta(S_A, T, P)$ and enthalpy $h(S_A, T, P)$ are now known and all the thermodynamic variables follow. That is, having formed enthalpy $h(S_A, T, P)$ from its partial derivatives (Eqn. 8) there is no need for an
additional definition; the Gibbs function and its definition do not need to be introduced. Rather, the two functions $h(S_A, T, P)$ and $\eta(S_A, T, P)$ can be regarded as a pair of functions that together define all the thermodynamic properties of seawater. In this alternative reasoning, the Gibbs function $g(S_A, T, P)$ is introduced as the last step, for the sole purpose that all the thermodynamic quantities can be derived from a single function rather than having to carry along the two separate functions $h(S_A, T, P)$ and $\eta(S_A, T, P)$. In this case, the adoption of the Gibbs function rather than using the two functions $h(S_A, T, P)$
and $\eta(S_A, T, P)$ serves the additional important service that the consistency requirement, $h_T = T\eta_T$, does not need to be separately enforced.

## 2.3 The case of $\widehat{h}(S_A, \Theta, P)$ and $\widehat{\eta}(S_A, \Theta)$

Now we consider the case of Conservative Temperature $\Theta$ taking the place of in situ temperature $T$ as the independent temperature variable. Appendix P of IOC et al., 2010 has shown that if expressions for both enthalpy and entropy are known

in the functional forms $\hat{h}(S_A, \Theta, P)$ and $\hat{\eta}(S_A, \Theta)$, this information is sufficient to derive all the thermodynamic quantities. This can be understood from realizing that $\eta = \hat{\eta}(S_A, \Theta)$ is equivalent to providing the implicit definition of $\Theta = \widehat{\Theta}(S_A, \eta)$ so that knowledge of $\hat{h}(S_A, \Theta, P)$ and $\hat{\eta}(S_A, \Theta)$ is equivalent to knowing $\widetilde{h}(S_A, \eta, P) = \hat{h}\left(S_A, \widehat{\Theta}(S_A, \eta), P\right)$, so that the three partial derivatives of $\widetilde{h}(S_A, \eta, P)$ can be written in terms of the partial derivatives of $\hat{h}(S_A, \Theta, P)$ and $\hat{\eta}(S_A, \Theta)$ (see Table 2 for these expressions). Since $\widetilde{h}(S_A, \eta, P)$ is a well-known and fundamental thermodynamic potential, this completes the discussion of why all thermodynamic properties can be found from knowledge of the two functions $\hat{h}(S_A, \Theta, P)$ and $\hat{\eta}(S_A, \Theta)$. IOC et al., (2010) stopped short of finding a single thermodynamic potential in terms of $(S_A, \Theta, P)$; this is done in the present paper.

There are two useful features that follow directly from the definition of Conservative Temperature as being proportional to potential enthalpy referenced to $P_0$, i.e. $c_p^0 \Theta \equiv \hat{h}(S_A, \Theta, P_0)$. The first feature is that that entropy has the functional form $\eta = \hat{\eta}(S_A, \Theta)$ and is not a function of pressure: this feature is due to Conservative Temperature possessing the "potential" property (as does both entropy and Absolute Salinity). The second feature is the very simple form of the first derivatives of enthalpy at $P_0$, namely that $\hat{h}_\Theta(S_A, \Theta, P_0) = c_p^0$ and $\hat{h}_{S_A}(S_A, \Theta, P_0) = 0$. Specific enthalpy is now defined in terms of $(S_A, \Theta, P)$ from its total differential,

$$\mathrm{d}h = \left(\mu + T\hat{\eta}_{S_A}\right)\mathrm{d}S_A + T\hat{\eta}_\Theta \mathrm{d}\Theta + v\mathrm{d}P, \tag{11}$$

which is simply a rearranged version of the Fundamental Thermodynamic Relation (FTR) in the form, $\mathrm{d}h - v\mathrm{d}P = \mu\mathrm{d}S_A + T\mathrm{d}\eta$, since $\mathrm{d}\eta = \hat{\eta}_{S_A}\mathrm{d}S_A + \hat{\eta}_\Theta \mathrm{d}\Theta$. Knowledge of $\hat{\mu}(S_A, \Theta, P)$, $\hat{T}(S_A, \Theta, P)$ and $\hat{v}(S_A, \Theta, P)$ are needed to find these partial derivatives in Eqn. (11), while $\hat{\eta}(S_A, \Theta)$ can be found from integrating the first two partial derivatives of Eqn. (11) evaluated at $P_0$, namely $0 = \hat{\mu}(S_A, \Theta, P_0) + (T_0 + \theta)\hat{\eta}_{S_A}$ and $c_p^0 = (T_0 + \theta)\hat{\eta}_\Theta$, where $(T_0 + \theta) = \hat{T}(S_A, \Theta, P_0)$, together with the arbitrary assignment $\hat{\eta}(S_{SO}, 0°C) = 0$. After having formed both $\hat{\eta}(S_A, \Theta)$ and $\hat{h}(S_A, \Theta, P)$ from the differential form Eqn. (11), we know from Appendix P of IOC et al. (2010) that all the thermodynamic variables of seawater follow, so that if one is willing to define seawater properties using these two functions, no more work is required. However it is convenient to define all the thermodynamic properties from a single thermodynamic potential function, and in this paper we have found such a function, $\hat{\phi}(S_A, \Theta, P)$, given by Eqn. (14) below, which contains the information of both $\hat{\eta}(S_A, \Theta)$ and $\hat{h}(S_A, \Theta, P)$ and from which these two functions can be found.

Note that in this $(S_A, \Theta, P)$ case, specific volume, $v = \hat{h}_P$, internal energy, $u = \hat{h} - P\hat{h}_P$, and the isentropic compressibility, $\kappa = -\hat{h}_{PP}/\hat{h}_P$, depend only on enthalpy, $\hat{h}(S_A, \Theta, P)$, and are independent of entropy, $\hat{\eta}(S_A, \Theta)$. This contrasts with the $(S_A, T, P)$ case where specific volume, $v = h_P - T\eta_P$, internal energy, $u = h - Ph_P + TP\eta_P$, and the isentropic compressibility, $\kappa = -(h_{PP}\eta_T - h_T\eta_{PP} + \eta_P^2)/(h_P\eta_T - h_T\eta_P)$, depend not only on enthalpy, $h(S_A, T, P)$, but also on entropy, $\eta(S_A, T, P)$. The simpler expressions for specific volume, internal energy, the isentropic compressibility and the sound speed in the $(S_A, \Theta, P)$ case compared with the $(S_A, T, P)$ case is also a feature of the $\widetilde{h}(S_A, \eta, P)$ thermodynamic potential and is due to the Conservative Temperature variable being a "potential" variable.

In the next section we introduce the new thermodynamic potential $\hat{\phi}(S_A, \Theta, P)$ and then compare its derivation and properties with the corresponding derivation and properties of the Gibbs function. This leads to a discussion of whether $\hat{\phi}(S_A, \Theta, P)$ is as thermodynamically fundamental as the Gibbs function $g(S_A, T, P)$.

## 3 A thermodynamic potential of seawater in terms of Conservative Temperature

### 3.1 Defining the thermodynamic potential $\widehat{\phi}(S_A, \Theta, P)$

Since Conservative Temperature $\Theta$ is the temperature variable that is recommended for use in marine science under TEOS-10 (taking the place of potential temperature $\theta$) it is of interest to determine if a thermodynamic potential of seawater can be found in terms of $\Theta$. From Appendix P of IOC et al. (2010), and section 2 above, we know that if we can find a single function from which enthalpy and entropy can be found in the functional forms $\hat{h}(S_A, \Theta, P)$ and $\hat{\eta}(S_A, \Theta)$, our aim will have been achieved. It is possible to find several such functions from which $\hat{h}(S_A, \Theta, P)$ and $\hat{\eta}(S_A, \Theta)$ can be derived, and some of these are described in Appendix A. The one we suggest, Eqn. (14) below, is motivated from section 5 of Feistel (2008) (the paper that derived the Gibbs function of seawater as incorporated into TEOS-10), where the differential expression for the Gibbs function, Eqn. (3), was integrated along an arbitrary but convenient path through $(S_A, T, P)$ space, first with respect to Absolute Salinity from the Absolute Salinity of Standard Seawater $S_{SO}$ at $T = T_0$ and $P = P_0$, then with respect to in situ temperature at the given Absolute Salinity and at $P = P_0$, and finally with respect to pressure at the given values of Absolute Salinity and in situ temperature, so that the Gibbs function can be written as

$$g(S_A, T, P) = \int_{S_{SO}}^{S_A} \mu(S_A', T_0, P_0) dS_A' - \int_{T_0}^{T} \eta(S_A, T', P_0) dT' + \int_{P_0}^{P} v(S_A, T, P') dP'. \tag{12}$$

where $g(S_{SO}, T_0, P_0)$ was chosen to be zero with no loss of generality. This integration method results in a path-independent function $g(S_A, T, P)$ if and only if the three integrands satisfy the integrability conditions (Maxwell relations) $\mu_T = -\eta_{S_A}$, $\mu_P = v_{S_A}$ and $-\eta_P = v_T$.

In this paper we adopt a similar integration of entropy and specific volume, but now with respect to Conservative Temperature (rather than in situ temperature) to define the new thermodynamic potential of seawater $\hat{\phi}(S_A, \Theta, P)$ as

$$\hat{\phi}(S_A, \Theta, P) = \int_{P_0}^{P} \hat{v}(S_A, \Theta, P') dP' - \int_{0}^{\Theta} \hat{\eta}(S_A, \Theta') d\Theta'. \tag{13}$$

or equivalently (since we know that $v = \hat{h}_P$ and $c_p^0 \Theta \equiv \hat{h}(S_A, \Theta, P_0)$)

$$\hat{\phi}(S_A, \Theta, P) \equiv \hat{h}(S_A, \Theta, P) - c_p^0 \Theta - \int_{0}^{\Theta} \hat{\eta}(S_A, \Theta') d\Theta'. \tag{14}$$

Note that (i), entropy $\hat{\eta}(S_A, \Theta)$ is not a function of pressure, and (ii), unlike in Eqn. (12), we find that in Eqn. (13) we do not need to perform a salinity integral of relative chemical potential $\mu$ in order to fully define the thermodynamic properties of seawater from $\hat{\phi}(S_A, \Theta, P)$. Expressions for $\hat{h}(S_A, \Theta, P)$ and $\hat{\eta}(S_A, \Theta)$ are obtained from $\hat{\phi}(S_A, \Theta, P)$ as follows,

$$\hat{h}(S_A, \Theta, P) = c_p^0 \Theta + \hat{\phi}(S_A, \Theta, P) - \hat{\phi}(S_A, \Theta, P_0) = c_p^0 \Theta + \int_{P_0}^{P} \hat{\phi}_P(S_A, \Theta, P')dP', \tag{15}$$


$$\hat{\eta}(S_A, \Theta) = -\hat{\phi}_\Theta(S_A, \Theta, P_0) = -\hat{\phi}_\Theta(S_A, \Theta, P) + \int_{P_0}^{P} \hat{\phi}_{P\Theta}(S_A, \Theta, P')dP', \tag{16}$$

and from Appendix P of IOC et al., 2010, we know that all the thermodynamic variables follow once we have expressions for $\hat{h}(S_A, \Theta, P)$ and $\hat{\eta}(S_A, \Theta)$. For example, the conversion formula of Conservative to in-situ temperature follows from $\hat{\phi}(S_A, \Theta, P)$ to be

$$\hat{T}(S_A, \Theta, P) = \frac{\hat{h}_\Theta}{\hat{\eta}_\Theta} = -\frac{c_p^0 + \hat{\phi}_\Theta(S_A, \Theta, P) - \hat{\phi}_\Theta(S_A, \Theta, P_0)}{\hat{\phi}_{\Theta\Theta}(S_A, \Theta, P_0)}. \tag{17}$$

Hence, we conclude that $\hat{\phi}(S_A, \Theta, P)$, defined by Eqn. (14), is a thermodynamic potential of seawater. The expressions for several thermodynamic variables in terms of $\hat{\phi}(S_A, \Theta, P)$ can be found in Appendix C.

In summary, we are using polynomial fits to entropy and enthalpy (or equivalently, specific volume), as functions of Conservative Temperature, knowing from Appendix P of IOC et al. 2010 that these fits in the forms $\hat{h}(S_A, \Theta, P)$ and $\hat{\eta}(S_A, \Theta)$ are sufficient to define all the thermodynamic variables of seawater. We have then found a way, Eqn. (14), to combine these

two polynomial functions into one function from which both $\hat{h}(S_A, \Theta, P)$ and $\hat{\eta}(S_A, \Theta)$ can be found.

## 3.2 Is the thermodynamic potential $\hat{\phi}(S_A, \Theta, P)$ equivalent to the Gibbs function?

In section 2 we suggested that internal energy expressed as a function of Absolute Salinity, entropy and specific volume is the most natural thermodynamic potential, but since mixing processes in the ocean occur at constant pressure rather than at constant volume, a more useful thermodynamic potential for seawater is enthalpy in the functional form $\overrightarrow{h}(S_A, \eta, P)$. Once

one knows that enthalpy is defined in terms of internal energy by $h \equiv u + Pv$, the FTR in its original form, $du + Pdv = Td\eta + \mu dS_A$, follows from $\overrightarrow{h}(S_A, \eta, P)$. Similarly, forming the Gibbs function from "observations" (that is, knowledge) of $\mu(S_A, T, P)$, $\eta(S_A, T, P)$ and $v(S_A, T, P)$, using the total differential $dg = \mu dS_A - \eta dT + vdP$ is not equivalent to the FTR since there is no link to the total differentials of either enthalpy or internal energy. Rather, to proceed from knowledge of the total differential of the Gibbs function to the FTR one needs the additional knowledge that $g \equiv u + Pv - T\eta$. The same result

for our new thermodynamic potential, $\hat{\phi}(S_A, \Theta, P)$, is proven in Appendix D, namely that knowledge of its definition, Eqn. (14), and its total derivative, Eqn. (C17), leads directly to the FTR.

In section 2 we introduced an alternate route to deriving the Gibbs function, using knowledge/observations of $\mu(S_A, T, P)$, $\eta(S_A, T, P)$ and $v(S_A, T, P)$ together with the differential form Eqn. (8) of enthalpy, $h \equiv u + Pv$, to find enthalpy in the form $h(S_A, T, P)$, which embodies the FTR. During this process the constraint $h_T = T\eta_T$ must be enforced. The combination of

the information in $h(S_A, T, P)$ and $\eta(S_A, T, P)$ serves to define all the thermodynamic quantities of seawater, and the FTR follows from Eqn. (8) without the need to introduce another function. The last step in this discussion of the Gibbs function is to introduce it as $g(S_A, T, P) = h(S_A, T, P) - T\eta(S_A, T, P)$ for the sole purpose that all the thermodynamic quantities can be derived from a single function.

Similarly, we showed in the $(S_A, \Theta, P)$ case that knowledge/observations of $\hat{\mu}(S_A, \Theta, P)$, $\hat{T}(S_A, \Theta, P)$ and $\hat{v}(S_A, \Theta, P)$

together with the differential form Eqn. (11) gives both enthalpy and entropy in the forms $\hat{h}(S_A, \Theta, P)$ and $\hat{\eta}(S_A, \Theta)$ which also embody the FTR. In this case however a single thermodynamic potential is not needed either to arrive at the FTR or to be able to derive all the thermodynamic quantities of seawater. In both the $(S_A, T, P)$ and $(S_A, \Theta, P)$ cases a single thermodynamic potential can be found; in one case as $g(S_A, T, P) = h(S_A, T, P) - T\eta(S_A, T, P)$ and in the other as (Eqn. 14), $\hat{\phi}(S_A, \Theta, P) = \hat{h}(S_A, \Theta, P) - c_p^0 \Theta - \int_0^\Theta \hat{\eta}(S_A, \Theta') d\Theta'$.

We conclude that the new thermodynamic potential $\hat{\phi}(S_A, \Theta, P)$ and the Gibbs function $g(S_A, T, P)$ are equivalent thermodynamic potentials of seawater. Both thermodynamic potentials are found from "observations" of, in one case $\mu, \eta$ and $v$, and in the other case $\mu, T$ and $v$, to constrain various derivatives of either $h(S_A, T, P)$ or $\hat{h}(S_A, \Theta, P)$, from which the FTR follows. All the thermodynamic properties of seawater can be derived from these expressions for enthalpy along with their corresponding expressions for entropy. Given these pairs of expressions for enthalpy and entropy, corresponding

thermodynamic potential functions can be found in the form of the Gibbs function or in the form of $\hat{\phi}(S_A, \Theta, P)$. This summarizes the identical nature of the derivations of the two thermodynamic potentials from the viewpoint of the slightly different derivation of the thermodynamic potentials as described in section 2. In Appendix D we describe the equivalence of these two potential functions on the basis of their differential expressions and their definitions.

Having argued that the two thermodynamic potentials, $g(S_A, T, P)$ and $\hat{\phi}(S_A, \Theta, P)$ are equivalent, we add a practical caveat

regarding how $\hat{\phi}(S_A, \Theta, P)$ has actually been found; that is, how we formed the polynomial expressions for $\hat{v}(S_A, \Theta, P)$ and $\hat{\eta}(S_A, \Theta)$, that appear in the definition of $\hat{\phi}(S_A, \Theta, P)$ in Eqn. (13). First, all the most accurate data of thermodynamic quantities (such as specific volume, sound speed, isobaric specific heat capacity, "heat of mixing", temperature of maximum density, freezing point depression, etc.) were absorbed into the TEOS-10 Gibbs function of seawater $g(S_A, T_0 + t, P)$ (Feistel 2003, 2008). It is natural to absorb this information into a Gibbs function because all the laboratory data were obtained at measured

values of in situ temperature, and the Gibbs function has in situ temperature as its "heat-like" independent variable. Second, the conversion between in situ and potential temperature used the implicit relationship $g_T(S_A, T_0 + t, P) = g_T(S_A, T_0 + \theta, P_0)$ which involves the Gibbs function. Third, the conversion between potential temperature and Conservative Temperature used the Gibbs function-based equation of potential enthalpy, $h(S_A, T_0 + \theta, P_0)$, which is equated to $c_p^0 \Theta$. Fourth, using this conversion between $t$ and $\Theta$ we were able, in Roquet et al. (2015), to form a polynomial expression for $\hat{v}(S_A, \Theta, P)$ from the

Gibbs function-based values of $v = g_P(S_A, T_0 + t, P)$. Fifth, and lastly, using the now known conversion between $\theta$ and $\Theta$,

we are able in this paper to form an algorithm for $\hat{\eta}(S_A, \Theta)$ from the Gibbs function-based values of $\tilde{\eta}(S_A, \theta) = -g_T(S_A, T_0 + \theta, P_0)$.

In summary, we have used the TEOS-10 Gibbs function of seawater to relate the different temperature variables and to evaluate both specific volume and entropy, which were then fitted with polynomials in the three independent variables $S_A, \Theta, P$.
In performing these polynomial fits, we ensured that in the oceanographic range of salinity, the $\hat{v}(S_A, \Theta, P)$ and $\hat{\eta}(S_A, \Theta)$ polynomials fitted the Gibbs function-derived values of specific volume and entropy more accurately than these variables are known from the underlying laboratory measurements. In this way we claim that the thermodynamic potential $\hat{\phi}(S_A, \Theta, P)$ and the TEOS-10 Gibbs function $g(S_A, T_0 + t, P)$ are equally accurate in representing the thermodynamic properties of seawater in the oceanographically relevant range of salinity.

## 4   An approximate polynomial expression for entropy

### 4.1 An analogy with a perfect gas

In order to construct an accurate polynomial expression for the thermodynamic potential of seawater $\hat{\phi}(S_A, \Theta, P)$ of Eqn. (13) we will integrate the 75-term polynomial expression for specific volume $\hat{v}(S_A, \Theta, P)$ of Roquet *et al.* (2015) with respect to pressure to obtain $\hat{h}(S_A, \Theta, P)$ (using the fact that $\hat{h}_P = v$), and we also need to find an accurate expression for entropy,
$\hat{\eta}(S_A, \Theta)$, which we will develop in this section.

    The specific entropy of a perfect gas can be expressed in terms of the Celsius potential temperature $\theta$ (with reference sea pressure of $p_r = 0$ dbar; that is, reference absolute pressure of $P_r = P_0 \equiv 101\,325$ Pa) by

$$\eta^{\text{gas}} = c_p^{\text{gas}} \ln(1 + \theta/T_0) \tag{18}$$

where entropy is defined so that it is zero at a Celsius temperature of 0°C (see Eqns. (J.6) and (J.7) of IOC *et al.* (2010)). In
general, the enthalpy and internal energy of a perfect gas is a general function of (only) temperature, but here we have restricted attention to the "calorically perfect gas" where the specific isobaric heat capacity $c_p^{\text{gas}}$ is a constant. The enthalpy of a perfect gas (e.g. dry air) is also defined to be zero at a Celsius temperature of 0°C, so the potential enthalpy of a perfect gas is $h^0 = c_p^{\text{gas}} \theta$ and if a "conservative temperature of a perfect gas" were to be defined, then it would be simply equal to potential temperature $\theta$.

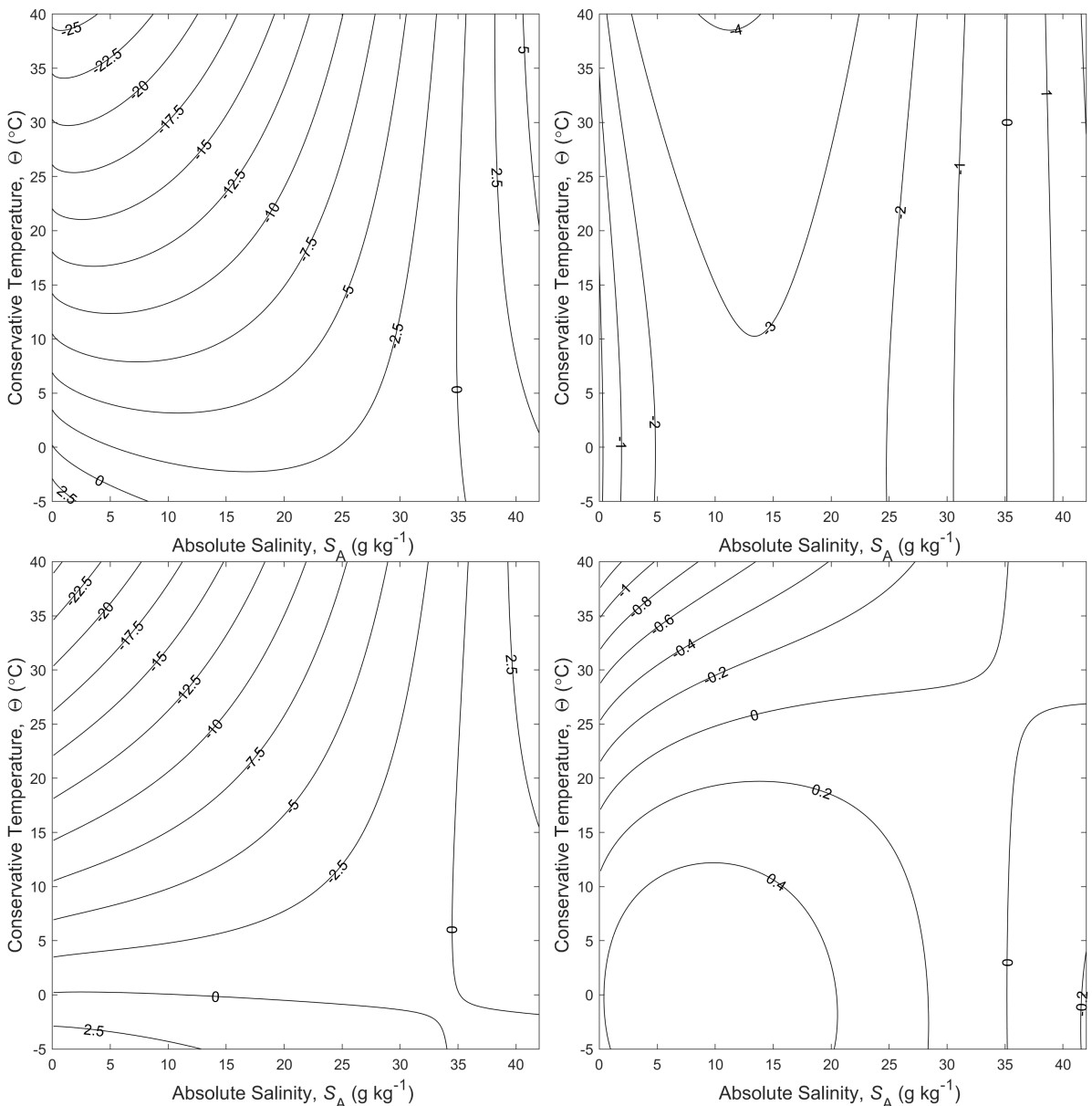


**Figure 1.** Panels (a) and (b) are contour plots of $c_p^0 \ln(1 + \theta/T_0) - \eta$ and $c_p^0 \ln(1 + \Theta/T_0) - \eta$ respectively, while panels (c) and (d) show $c_p^0 \ln(1 + \theta/T_0) + a(S_A/S_{SO}) \ln(S_A/S_{SO}) - \eta$ and $c_p^0 \ln(1 + \Theta/T_0) + a(S_A/S_{SO}) \ln(S_A/S_{SO}) - \eta$ respectively. All panels in this figure are in the units of entropy, namely $J \, kg^{-1} \, K^{-1}$.

One wonders how accurate a correspondingly simple logarithm expression would be for the entropy of seawater, defined by either $c_p^0 \ln(T_0 + \theta) + \text{constant}$ or by $c_p^0 \ln(T_0 + \Theta) + \text{constant}$. The constants can be chosen so that it makes the estimate of entropy zero if $\theta = 0°C$ or $\Theta = 0°C$ in the two cases respectively, since entropy is zero for Standard Seawater ($S_A = S_{SO}$) at this temperature. That is, we examine the two estimates $c_p^0 \ln(1 + \theta/T_0)$ and $c_p^0 \ln(1 + \Theta/T_0)$ as approximations to the entropy of seawater. (Note that for seawater, $c_p^0$ is approximately four times as large as the isobaric specific heat capacity of air, $c_p^{\text{gas}}$). The errors in using these two approximate expressions can be seen in Figure 1 (a) and (b). It is seen that the expression involving Conservative Temperature, $c_p^0 \ln(1 + \Theta/T_0)$, is a better approximation to entropy than is the one involving potential temperature, $c_p^0 \ln(1 + \theta/T_0)$, with the maximum error being less by approximately an order of magnitude. The relative accuracies of these approximate expressions to the specific entropy of seawater can be understood from the following expressions for the total differential of entropy in terms of $\theta$ and $\Theta$ (see Eqns. (A.12.7) and (A.12.8) of IOC et al. 2010),

$$d\eta = c_p(S_A, \theta, P_0) \, d(\ln[1 + \theta/T_0]) - \mu_T(S_A, \theta, P_0) \, dS_A, \tag{19}$$

$$d\eta = c_p^0 \frac{(T_0+\Theta)}{(T_0+\theta)} \, d(\ln[1 + \Theta/T_0]) - \frac{\hat{\mu}(S_A,\Theta,P_0)}{(T_0+\theta)} \, dS_A. \tag{20}$$

The partial derivative with respect to Absolute Salinity that has been used in Eqn. (19), namely $\tilde{\eta}_{S_A} = \eta_{S_A}(S_A, \theta, P_0)$, is also given by $-\mu_T(S_A, \theta, P_0)$ since both expressions are $-g_{TS_A}(S_A, \theta, P_0)$, while the other partial derivative, $\tilde{\eta}_\theta = c_p(S_A, \theta, P_0)/(T_0 + \theta)$, can be gleaned from $h_T = T\eta_T$ (from Eqn. 8) evaluated at $P_0$, noting that $c_p(S_A, \theta, P_0) = h_T(S_A, \theta, P_0)$ is the specific isobaric heat capacity of seawater evaluated at $P_0$ and at the potential temperature $\theta$. The partial derivatives $\hat{\eta}_{S_A}$ and $\hat{\eta}_\Theta$ used in Eqn. (20) can be gleaned from Eqn. (11) evaluated at $P_0$, noting that $\hat{h}_{S_A}(S_A, \Theta, P_0) = 0$. The contributions of the terms in $dS_A$ are small in comparison to the leading terms on the right-hand sides of Eqns. (19) and (20), and the specific heat capacity $c_p(S_A, \theta, P_0)$ varies by 5.5% in the ocean whereas the ratio $(T_0 + \Theta)/(T_0 + \theta)$ varies by no more than 0.67%, and this explains why the approximate expression $\eta \approx c_p^0 \ln(1 + \Theta/T_0)$ out-performs $\eta \approx c_p^0 \ln(1 + \theta/T_0)$ by about an order of magnitude.

While the fit to entropy in better in Fig. 1(b) than in Fig. 1(a), neither is particularly accurate for our purposes. For example, in determining potential temperature $\theta$ from $\hat{\eta}_\Theta = c_p^0/(T_0 + \theta)$, the remaining error in Fig. 1(b) amounts to an error in potential temperature of approximately 0.5°C while that in Fig 1(a), using $\tilde{\eta}_\theta = c_p(S_A, \theta, P_0)/(T_0 + \theta)$, amounts to about 10°C.

## 4.2 Adding a simple function of Absolute Salinity

The Second Law of Thermodynamics requires that entropy must be produced when mixing occurs, and the approximation $c_p^0 \ln(1 + \Theta/T_0)$ does not allow for the production of entropy when mixing occurs between seawater parcels of different

Absolute Salinities but the same value of Conservative Temperature. The TEOS-10 Gibbs-function-derived expression for specific entropy contains the term $a(S_A/S_{SO})\ln(S_A/S_{SO})$ with the coefficient $a$ being $a =$

$-9.310\,292\,413\,479\,596\,\text{J kg}^{-1}\,\text{K}^{-1}$ (this is the value of the coefficient derived from the $g_{110}$ coefficient of the Gibbs function (appendix H of IOC *et al.* (2010)), allowing for our version of the normalization of salinity, $(S_A/S_{SO})$). This term was derived by Feistel (2008) to be theoretically correct at very small Absolute Salinities, relying on Plank's theory of ideal solutions and the now-exact value of the molar gas constant. Here we use the slightly different value $a =$ $-9.309\,495\,003\,228\,781\,J\,\text{kg}^{-1}\,\text{K}^{-1}$ that comes from a least-squares fit incorporating a particular polynomial form, as

described below, and tabulated in appendix B. This slightly different value of $a$ allows a more accurate fit to the entropy data over the whole range of oceanographic salinities rather than only at vanishingly small salinities.

The accuracy of the approximate expression $c_p^0\ln(1+\Theta/T_0)+a(S_A/S_{SO})\ln(S_A/S_{SO})$ is shown in Figure 1(d). There is no improvement over $c_p^0\ln(1+\Theta/T_0)$ near zero Absolute Salinity, but at intermediate salinity values the fit is improved over that of $c_p^0\ln(1+\Theta/T_0)$ by approximately an order of magnitude. Over the whole $(S_A,\Theta)$ plane this simple theoretically

inspired estimate of entropy, illustrated in Figure 1(d) is in error by no more than 0.2% of the full range of entropy. In contrast, when the same expression is used with potential temperature (see Figure 1(c)) in place of Conservative Temperature, the relative error is 4% of the full range of entropy.

### 4.3 The full expression for $\hat{\eta}(S_A,\Theta)$

In order to obtain an expression for $\hat{\eta}(S_A,\Theta)$ suitable for combining with the 75-term polynomial for specific volume

$\hat{v}(S_A,\Theta,P)$ of Roquet *et al.* (2015) to form the thermodynamic potential of seawater $\hat{\phi}(S_A,\Theta,P)$ of Eqn. (13), we have added a polynomial in powers of $s=[S_A/S_{SO}]^{0.5}$ and $\tau=\Theta/40°C$ with the highest power of each being eight, so that our final approximate expression for $\hat{\eta}(S_A,\Theta)$ is

$$\hat{\eta}(S_A,\Theta) = c_p^0\ln(1+\Theta/T_0)+a(S_A/S_{SO})\ln(S_A/S_{SO})+P\{8,8\}(s,\tau), \tag{21}$$

and the 45 coefficients of the eighth order bi-polynomial $P\{8,8\}$ are listed in Appendix B. The error of Eqn. (21) in

approximating $\hat{\eta}(S_A,\Theta)$ is shown in Fig. 2(a), from which we see that the typical error is $2\times10^{-6}\,\text{J kg}^{-1}\,\text{K}^{-1}$.

When the thermodynamic potential $\hat{\phi}(S_A,\Theta,P)$ of Eqn. (13 or 14) is used to obtain all the thermodynamic properties of seawater, one of the key variables that is obtained from entropy in the form $\hat{\eta}(S_A,\Theta)$ is the potential temperature $\theta$ referenced to $P_0$, and this is found from the derivative of entropy with respect to Conservative Temperature, namely

$$\hat{\eta}_\Theta = \frac{c_p^0}{(T_0+\theta)}. \tag{22}$$

This relationship was originally derived from the FTR by McDougall (2003) and can be deduced from Eqn. (11); see also Eqn. (A.12.8) of the TEOS-10 Manual (IOC *et al.* (2010)). When the polynomial-based approximate form of $\hat{\eta}(S_A,\Theta)$, Eqn. (21), is used to evaluate the potential temperature from Eqn. (22), the error is as shown in Fig. 2(b), where we see that the typical

error is $10\mu K$, with maximum errors of $60\mu K$ at $S_A = 0 \, \text{g kg}^{-1}$. Since this error seems acceptable in oceanographic applications, and since the 75-term polynomial for $\hat{v}(S_A, \Theta, P)$ of Roquet *et al.* (2015) is as accurate in the oceanographic range

of salinity as the data to which the original Gibbs function of Feistel (2008) was fitted, we conclude that the thermodynamic potential $\hat{\phi}(S_A, \Theta, P)$ of Eqn. (13 or 14), which is written in terms of Conservative Temperature, is equally as accurate as the Gibbs function $g(S_A, T, P)$, and will therefore prove sufficiently accurate for use in physical oceanography as the thermodynamic potential of seawater in the oceanographic range of salinity.

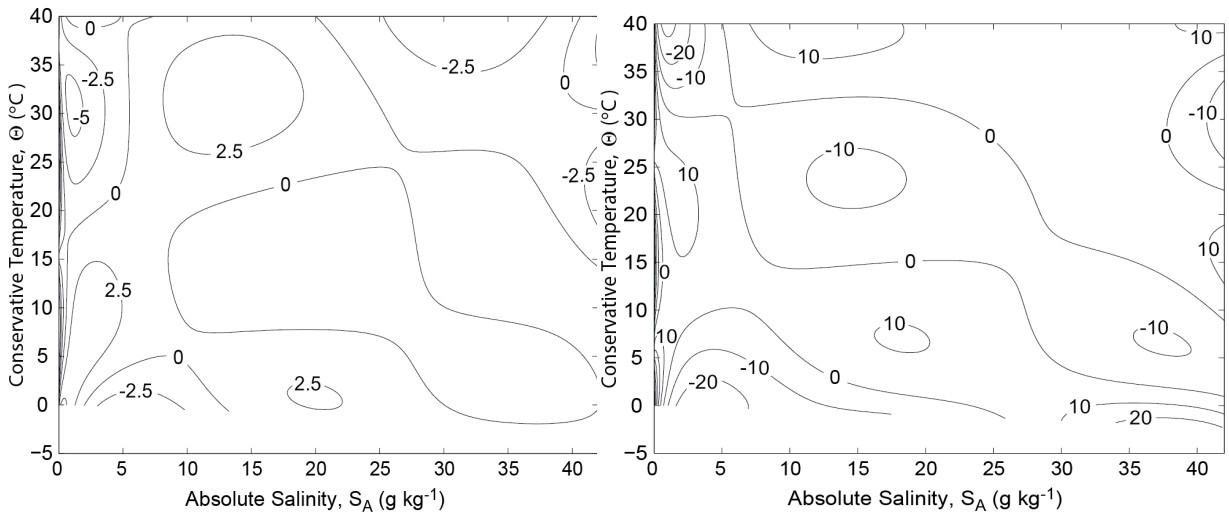


**Figure 2.** (a) The error in the fit Eqn. (21) to entropy (in units of $10^{-6} \, \text{J kg}^{-1} \, \text{K}^{-1}$). (b) The error in evaluating potential temperature $\theta$ (in $\mu K$) from Eqns. (21) and (22).

## 5 Numerical Implementation

When calculating Conservative Temperature $\Theta$ from observations of in situ temperature $t$ using the Gibbs function approach, the first step is to calculate the potential temperature $\theta$ at the reference pressure $P_0$ by equating the values of entropy at the in situ pressure $P$ and at the reference pressure $P_0$, that is, by solving the implicit relationship $g_T(S_A, T_0 + t, P) = g_T(S_A, T_0 + \theta, P_0)$. The second step is to evaluate the parcel's potential enthalpy, $h(S_A, T_0 + \theta, P_0)$, being $g(S_A, T_0 + \theta, P_0) - (T_0 + \theta)g_T(S_A, T_0 + \theta, P_0)$, and the third step is to divide potential enthalpy by $c_p^0$. The computationally expensive step is the

first, typically involving a Newton-type iterative procedure.

When adopting the approach of the present paper, the conversion from in situ temperature $t$ to Conservative Temperature $\Theta$ is also computationally expensive, since, from Eqn. (11), $\Theta$ is obtained by finding the zero of the function $\hat{h}_\Theta / \hat{\eta}_\Theta - (T_0 + t)$. This is done by first evaluating both an approximate polynomial for $\Theta$ as a function of $(S_A, T_0 + t, P)$, and an approximation to the second derivative of $\Theta$ with respect to in situ temperature, by differentiating the polynomial. Then only one pass though the accelerated Newton method of McDougall et al. (2019) is needed to evaluate $\Theta$ to machine precision. This code takes approximately the same time to compute $\Theta$ as does using the Gibbs function approach described in the previous paragraph.

Having converted observations of in situ temperature into Conservative Temperature, other calculations are more computationally efficient when using the enthalpy and entropy combination of $\hat{h}(S_A, \Theta, P)$ and $\hat{\eta}(S_A, \Theta)$ of the present paper rather than the Gibbs function $g(S_A, T, P)$. For example, during the running of an ocean model, the sea surface temperature is needed as the input temperature for bulk air-sea flux formulae. With the approach of the present paper this is a forward calculation requiring only the evaluation of $\hat{\eta}_\Theta$ since in this case the sea surface temperature, $\theta$, is given by the simple forward expression $(T_0 + \theta) = c_p^0 / \hat{\eta}_\Theta$. This calculation is a factor of three less computationally expensive than the corresponding calculation based on the Gibbs function (where an iterative Newton-based algorithm is required).

Similar gains in computational efficiency occur when evaluating potential density at a variety of reference pressures when using $\hat{v}(S_A, \Theta, P)$ compared with the Gibbs function approach. These computational gains occur because the potential specific volume, referenced to an arbitrary reference pressure $P_r$, is available from the forward polynomial expression $\hat{v}(S_A, \Theta, P_r)$, whereas with the Gibbs function approach, the potential temperature referenced to $P_r$ needs to be evaluated and this involves an iterative calculation.

We have written algorithms to evaluate all of the thermodynamic quantities of seawater using only one or both of $\hat{h}(S_A, \Theta, P)$ and $\hat{\eta}(S_A, \Theta)$, and we have not found the need to use the new thermodynamic potential $\hat{\phi}(S_A, \Theta, P)$ itself nor any of its derivatives. That is, the most direct and computationally efficient way of adopting the approach of this paper was found to be to use $\hat{h}(S_A, \Theta, P)$ and $\hat{\eta}(S_A, \Theta)$ rather than their parent function $\hat{\phi}(S_A, \Theta, P)$. This is because the thermodynamic information in $\hat{h}(S_A, \Theta, P)$ is independent of that in $\hat{\eta}(S_A, \Theta)$, and would suggest that the fact that we have found a thermodynamic potential for seawater in the functional form, $\hat{\phi}(S_A, \Theta, P)$, is of theoretical thermodynamic interest, but so far has not yielded a practical benefit.

**6 Conclusions**

While in situ temperature is relatively simple to measure in the ocean, it is neither a "potential" property, nor is it a "conservative" property, and these deficiencies of in situ temperature have led to the adoption of Conservative Temperature $\Theta$ for use in physical oceanography. This switch to Conservative Temperature since the introduction of TEOS-10 in 2010, has motivated the quest of this paper: to find a thermodynamic potential of seawater in terms of Conservative Temperature, Absolute Salinity and pressure. Roquet et al. (2015) have provided a 75-term polynomial for specific volume in the form

$\hat{v}(S_A, \Theta, P)$ and this is the basis for many of the functions in the Gibbs Seawater (GSW) Oceanographic Toolbox of TEOS-10. But to date the conversions between in situ temperature and Conservative Temperature have been done using the TEOS-10 Gibbs function, and this is not 100% consistent with the use of the Roquet et al. polynomial for $\hat{v}(S_A, \Theta, P)$.

When the Roquet et al. (2015) 75-term polynomial for specific volume, $\hat{v}(S_A, \Theta, P)$, is integrated with respect to pressure (noting that $\hat{v} = \hat{h}_P$) and the resulting polynomial for enthalpy is used in the expression for the ratio of the in situ and potential temperatures, $(T_0 + t)/(T_0 + \theta) = \hat{h}_\Theta(S_A, \Theta, P)/c_p^0$, the difference between these temperatures, $|t - \theta|$, compared with evaluating this temperature difference using the Gibbs function, is not quite zero, with typical values being less than $10^{-4}$K (the standard deviation of the temperature difference is $4 \times 10^{-5}$K; see Table 3 of Roquet et al., 2015). From Figure 2(b) above

we see that the use of the $\hat{\eta}(S_A, \Theta)$ expression of the present paper has errors when relating potential and Conservative temperatures of $10^{-5}$K. The sum of these tiny temperature differences amounts to less than $10^{-4}$K, representing the difference in evaluating Conservative Temperature from in situ temperature using the Gibbs function versus using the Roquet et al. (2015) expression for $\hat{v}(S_A, \Theta, P)$ together with the present expression for $\hat{\eta}(S_A, \Theta)$. These inconsistencies in temperature between the two approaches are small, being more than an order of magnitude smaller than the underlying experimental error in the

laboratory data from which the TEOS-10 Gibbs function was derived. While these differences are small, it is preferable if all the thermodynamic quantities are 100% consistent with each other, and the approach presented in the present paper ensures this.

        In this paper we have provided an accurate expression for entropy as a function of Conservative Temperature, $\hat{\eta}(S_A, \Theta)$, and this can be used in conjunction with Roquet et al.'s $\hat{v}(S_A, \Theta, P)$ to relate in situ temperature and Conservative Temperature.

These relationships between the different temperature variables can be performed consistently, to machine precision, and without further reference to the Gibbs function $g(S_A, T, P)$. Appendix P of IOC et al. (2010) has shown that knowledge of both enthalpy and entropy in the functional forms $\hat{h}(S_A, \Theta, P)$ and $\hat{\eta}(S_A, \Theta)$ is sufficient to derive all thermodynamic variables, so it seems advisable that when the 75-term polynomial of Roquet et al. 2015 is used, that it is used in conjunction with the expression for $\hat{\eta}(S_A, \Theta)$ of the present paper. When the results of the present paper are implemented in the Gibbs SeaWater

(GSW) Oceanographic Toolbox of TEOS-10 (McDougall and Barker, 2011), the functions that will be changed are (i) those that calculate one of $\eta, \Theta, \theta, T$ from another one, (ii) the adiabatic lapse rate, (iii) the calculation of the three chemical potentials and the Gibbs function, as well as (iv) the provision of the new thermodynamic potential $\hat{\phi}(S_A, \Theta, P)$.

        Converting from observed values of in situ temperature to Conservative Temperature takes a similar amount of computer time using the $(S_A, \Theta, P)$ approach of the present paper as when using the Gibbs function, but the subsequent calculations of

various temperatures and potential densities are more computationally efficient using the $(S_A, \Theta, P)$ approach since these quantities require only simple forward (as opposed to iterative) calculations.

        In the $(S_A, \Theta, P)$ case, specific volume, internal energy, the isentropic compressibility and the sound speed depend only on enthalpy, $\hat{h}(S_A, \Theta, P)$, and are independent of entropy, $\hat{\eta}(S_A, \Theta)$, whereas the expressions for the corresponding variables in the $(S_A, T, P)$ case depend not only on enthalpy, $h(S_A, T, P)$, but also on entropy, $\eta(S_A, T, P)$. In the $(S_A, \Theta, P)$ case, the additional

information in $\hat{\eta}(S_A, \Theta)$ is needed to switch between the "temperature-like" variables $\eta, \theta, T, \Theta$ and to evaluate the chemical potentials. Thus the $\hat{h}(S_A, \Theta, P)$ and $\hat{\eta}(S_A, \Theta)$ pair provides a clean separation of the heat and buoyancy information (derivable from $\hat{h}(S_A, \Theta, P)$ alone) from the information in $\hat{\eta}(S_A, \Theta)$ that is needed to relate the various temperature variables and the chemical potentials. Also, unlike in the $(S_A, T, P)$ case, there is no consistency requirement between $\hat{h}(S_A, \Theta, P)$ and $\hat{\eta}(S_A, \Theta)$.

     Moreover, we have been able to combine the expressions for specific volume and for entropy into a single thermodynamic

potential function, $\hat{\phi}(S_A, \Theta, P)$, Eqn. (14), repeated here,

$$\hat{\phi}(S_A, \Theta, P) = \hat{h}(S_A, \Theta, P) - \int_0^\Theta \left[c_p^0 + \hat{\eta}(S_A, \Theta')\right]d\Theta', \tag{23}$$

$$= \int_{P_0}^P \hat{v}(S_A, \Theta, P')dP' - \int_0^\Theta \hat{\eta}(S_A, \Theta')d\Theta'.$$

from which all the thermodynamic quantities of seawater can be derived (see Appendix C). The clean separation of the various thermodynamic information between the $\hat{h}(S_A, \Theta, P)$ and $\hat{\eta}(S_A, \Theta)$ pair of functions, as well as the new-found ability to form

the thermodynamic potential $\hat{\phi}(S_A, \Theta, P)$ perhaps provides a theoretical boost to using Conservative Temperature as the temperature variable in physical oceanography as recommended by TEOS-10 (Valladares et al., 2011a,b). The thermodynamic potential, $\hat{\phi}(S_A, \Theta, P)$, is both complete (in that every thermodynamic property can be derived from it), and consistent (in that there is only one expression for each thermodynamic quantity). As an aside, we mention that we have also been able to find a thermodynamic potential of seawater, $\tilde{\psi}(S_A, \theta, P)$, in terms of potential temperature $\theta$, Eqn. (A8), repeated here,


$$\tilde{\psi}(S_A, \theta, P) \equiv \tilde{h}(S_A, \theta, P) - [T_0 + \theta]\,\tilde{\eta}(S_A, \theta), \tag{24}$$

$$= \int_{P_0}^P \tilde{v}(S_A, \theta, P')dP' + g(S_A, \theta, P_0),$$

$$= \int_{P_0}^P \tilde{v}(S_A, \theta, P')dP' - \int_0^\theta \tilde{\eta}(S_A, \theta')d\theta' + g(S_A, 0°C, P_0).$$

We note in passing that the first term on the last lines of Eqns. (23) and (24), namely the pressure integral terms, are equal to each other, while the second terms, the temperature integral terms, are not the same. The last line of Eqn. (24) has the same

form as Eqn. (12), but now with $\theta$ replacing $T - T_0$.

     The fact that we have been able to form the new thermodynamic potential, $\hat{\phi}(S_A, \Theta, P)$, is perhaps less important than the key insight of Appendix P of IOC et al. (2010) that knowledge of both $\hat{h}(S_A, \Theta, P)$ and $\hat{\eta}(S_A, \Theta)$ is sufficient to describe all the thermodynamic properties of seawater. We now better appreciate this insight, and also the importance of the fact that the thermodynamic information in $\hat{h}(S_A, \Theta, P)$ is completely independent of that in $\hat{\eta}(S_A, \Theta)$. While finding a thermodynamic

potential for seawater in the functional form $\hat{\phi}(S_A, \Theta, P)$ is of theoretical thermodynamic interest, so far this has not yielded a practical benefit that exceeds the knowledge of both $\hat{h}(S_A, \Theta, P)$ and $\hat{\eta}(S_A, \Theta)$ individually. That is, these two functions, together, act like a thermodynamic potential, and we have not actually found a pressing need to combine them into a single function, even though we have been able to do so. By contrast, the thermodynamic information in enthalpy is *not independent* of that in entropy when expressed as functions of $(S_A, \theta, P)$, so that finding the thermodynamic potential, $\tilde{\psi}(S_A, \theta, P)$, in terms

of potential temperature does add value because its use automatically enforces the consistency requirement $\tilde{h}_\theta(S_A, \theta, P_0) = [T_0 + \theta]\tilde{\eta}_\theta(S_A, \theta)$.

    The thermodynamic independence of the information contained in $\hat{h}(S_A, \Theta, P)$ and $\hat{\eta}(S_A, \Theta)$ appears to be a unique feature, due to the use of Conservative Temperature. For example, when changing thermodynamic potentials from $u(S_A, \eta, v)$ to $h(S_A, \eta, P) = u(S_A, \eta, P) + Pv(S_A, \eta, P)$, one cannot simply use the combination of $u(S_A, \eta, P)$ and $v(S_A, \eta, P)$ as effectively

a thermodynamic potential without first imposing the consistency constraint $u_P(S_A, \eta, P) = -Pv_P(S_A, \eta, P)$. Similarly when forming the Helmholtz function by $f(S_A, T, v) = u(S_A, T, v) - T\eta(S_A, T, v)$, one cannot simply use $u(S_A, T, v)$ together with $\eta(S_A, T, v)$ as effectively a thermodynamic potential without first imposing the constraint $u_T(S_A, T, v) = T\eta_T(S_A, T, v)$. So, among all the thermodynamic potential functions in common use, it is only when the "temperature-like" variable is Conservative Temperature that the thermodynamic information in enthalpy is independent of that in entropy so that the

combined knowledge of $\hat{h}(S_A, \Theta, P)$ and $\hat{\eta}(S_A, \Theta)$ can be used to define all the thermodynamic properties of seawater without the need of an additional consistency constraint.

**Appendix A: Alternative thermodynamic potentials in terms of Conservative Temperature and potential temperature**

Eqn. (14) (or Eqn. (13)) above is the proposed definition of the thermodynamic potential of seawater defined with respect to Conservative Temperature, but it is not the only possible functional form, and here we present other possibilities. Eqns. (13) and (14) resemble the integral definition of the Gibbs function, Eqn. (12), and now we follow an analogy with the $g = h - T\eta$ definition of the Gibbs function by considering the following three possible thermodynamic potentials

$$\hat{\varphi}(S_A, \Theta, P) \equiv \hat{h}(S_A, \Theta, P) - T_0\,\hat{\eta}(S_A, \Theta), \tag{A1}$$


$$\hat{\varpi}(S_A, \Theta, P) \equiv \hat{h}(S_A, \Theta, P) - [T_0 + \Theta]\hat{\eta}(S_A, \Theta), \tag{A2}$$

$$\hat{\pi}(S_A, \Theta, P) \equiv \hat{h}(S_A, \Theta, P) - c_p^0 \hat{\eta}(S_A, \Theta)/\hat{\eta}_\Theta(S_A, \Theta), \tag{A3}$$

where in the last equation $c_p^0/\hat{\eta}_\Theta(S_A, \Theta)$ is another way of writing $[T_0 + \theta]$. In each of these cases enthalpy can be found using the same functional form as Eqn. (15), that is,

$$
\begin{aligned}
\hat{h}(S_A, \Theta, P) &= c_p^0\Theta + \hat{\varphi}(S_A, \Theta, P) - \hat{\varphi}(S_A, \Theta, P_0) &= c_p^0\Theta + \int_{P_0}^{P} \hat{\varphi}_P(S_A, \Theta, P')\mathrm{d}P', \\
&= c_p^0\Theta + \hat{\varpi}(S_A, \Theta, P) - \hat{\varpi}(S_A, \Theta, P_0) &= c_p^0\Theta + \int_{P_0}^{P} \hat{\varpi}_P(S_A, \Theta, P')\mathrm{d}P', \\
&= c_p^0\Theta + \hat{\pi}(S_A, \Theta, P) - \hat{\pi}(S_A, \Theta, P_0) &= c_p^0\Theta + \int_{P_0}^{P} \hat{\pi}_P(S_A, \Theta, P')\mathrm{d}P'.
\end{aligned}
\tag{A4}
$$


In the case of $\hat{\varphi}(S_A, \Theta, P)$, entropy is readily found from,

$$\hat{\eta}(S_A, \Theta) = \left[c_p^0\Theta - \hat{\varphi}(S_A, \Theta, P_0)\right]/T_0. \tag{A5}$$

In the case of $\hat{\varpi}(S_A, \Theta, P)$, values of entropy can be evaluated from


$$\hat{\eta}(S_A, \Theta) = \left[c_p^0\Theta - \hat{\varpi}(S_A, \Theta, P_0)\right]/[T_0 + \Theta], \tag{A6}$$

but to obtain a functional expression (for example, a polynomial) for $\hat{\eta}(S_A, \Theta)$ requires equating powers of $S_A$ and $\Theta$ between $[T_0 + \Theta]\hat{\eta}(S_A, \Theta)$ and $\left[c_p^0\Theta - \hat{\varpi}(S_A, \Theta, P_0)\right]$. This is quite possible but is a little less convenient than using Eqn. (16) or (A5).

In the case of $\hat{\pi}(S_A, \Theta, P)$, entropy must obey the differential equation,

$$\hat{\eta}_\Theta(S_A, \Theta)\left[\hat{\pi}(S_A, \Theta, P_0) - c_p^0\Theta\right] + c_p^0\hat{\eta}(S_A, \Theta) = 0, \tag{A7}$$

whose solution is not straightforward. For example, if $\hat{\pi}(S_A, \Theta, P_0)$ were a polynomial in $S_A$ and $\Theta$, then $\hat{\eta}(S_A, \Theta)$ cannot be a polynomial because if it were, the powers of $\Theta$ would be unbalanced in Eqn. (A7).

We conclude that both Eqns. (14) and (A1) are straightforward to use as thermodynamic potentials in terms of $(S_A, \Theta, P)$, while, with a little more effort, Eqn. (A2) can also be made to work. However, Eqn. (A3), whose right-hand side can be expressed as $\hat{h}(S_A, \Theta, P) - [T_0 + \theta]\hat{\eta}(S_A, \Theta)$, is unworkable. We have a slight preference for Eqn. (14) over Eqns. (A1) or

(A2), because when using (14), entropy arises as a temperature derivative of the thermodynamic potential, as it does for the Gibbs function, whereas in (A5) and (A6) entropy is proportional to the difference between $c_p^0\Theta$ and the thermodynamic potential.

We note that the functional form of Eqn. (A2) also works as a thermodynamic potential when potential temperature $\theta$ is used in place of $\Theta$ as the "heat-like" variable, with the caveat that the $\tilde{\eta}(S_A, \theta)$ and $\tilde{h}(S_A, \theta, P)$ functions that are used to construct this thermodynamic potential, (A8), need to satisfy the consistency relationship $\tilde{h}_\theta(S_A, \theta, P_0) \equiv [T_0 + \theta]\tilde{\eta}_\theta(S_A, \theta)$. This thermodynamic potential is

$$\tilde{\psi}(S_A, \theta, P) \equiv \tilde{h}(S_A, \theta, P) - [T_0 + \theta]\,\tilde{\eta}(S_A, \theta), \tag{A8}$$

$$= \int_{P_0}^{P} \tilde{v}(S_A, \theta, P')dP' + g(S_A, \theta, P_0),$$

$$= \int_{P_0}^{P} \tilde{v}(S_A, \theta, P')dP' - \int_0^\theta \tilde{\eta}(S_A, \theta')d\theta' + g(S_A, 0°\text{C}, P_0).$$

with the expressions for $\tilde{h}(S_A, \theta, P)$ and $\tilde{\eta}(S_A, \theta)$ being

$$\tilde{h}(S_A, \theta, P) = \tilde{\psi}(S_A, \theta, P) - [T_0 + \theta]\,\tilde{\psi}_\theta(S_A, \theta, P_0), \tag{A9}$$

$$\tilde{\eta}(S_A, \theta) = -\tilde{\psi}_\theta(S_A, \theta, P_0). \tag{A10}$$

These expressions for enthalpy and entropy are similar to the corresponding expressions in terms of the Gibbs function, with the difference being that entropy here is evaluated at $P_0$ rather than at the in-situ pressure $P$ (this last observation also applies to Eqn. (16)). If we restrict attention to processes occurring at $P_0$, the thermodynamic potential (A8) is a Legendre transformation of the thermodynamic potential $\overset{\frown}{h}(S_A, \eta, P)$. At this pressure, (A8) is the Gibbs function. The other thermodynamic potentials discussed in this paper (as well as (A8) at pressures other than $P_0$) are not the result of Legendre transformations. The second line of Eqn. (A8) has exploited the fact that the pressure derivative of $\tilde{h}(S_A, \theta, P)$ is specific volume. In this form of Eqn. (A8) a new polynomial expression for $\tilde{\eta}(S_A, \theta)$ is not required. Rather the Gibbs function itself is used, along with a polynomial expression for $\tilde{v}(S_A, \theta, P)$.

Of the other functional forms we have used above, namely Eqns. (14), (A1) and (A3), the only other viable form we have found as a function of $(S_A, \theta, P)$ is one that is similar to (14), being

$$\hat{\vartheta}(S_A, \theta, P) \equiv \tilde{h}(S_A, \theta, P) - \int_0^\theta \tilde{\eta}(S_A, \theta')d\theta'. \tag{A11}$$

Differentiating Eqn. (A11) with respect to potential temperature at $P_0$, and using $\tilde{h}_\theta(S_A, \theta, P_0) = [T_0 + \theta]\tilde{\eta}_\theta(S_A, \theta)$, it is found that entropy must obey the differential equation,

$$\hat{\vartheta}_\theta(S_A, \theta, P_0) = [T_0 + \theta]\tilde{\eta}_\theta(S_A, \theta) - \tilde{\eta}(S_A, \theta). \tag{A12}$$

To obtain a functional expression (for example, a polynomial) for $\tilde{\eta}(S_A, \theta)$ requires equating powers of $S_A$ and $\theta$ in Eqn. (A12). This is quite possible but is less convenient than using Eqn. (10). Having found $\tilde{\eta}(S_A, \theta)$, this can be integrated with respect to $\theta$ and used, together with $\hat{\vartheta}(S_A, \theta, P)$, to find enthalpy, $\tilde{h}(S_A, \theta, P)$, from Eqn. (A11).

If one did want to express all the thermodynamic variables in terms of $(S_A, \theta, P)$, a thermodynamic potential such as Eqn. (A8) is required (as opposed to using the two separate functions $\tilde{h}(S_A, \theta, P)$ and $\tilde{\eta}(S_A, \theta)$) because the use of the thermodynamic potential ensures that the consistency relationship $\tilde{h}_\theta(S_A, \theta, P_0) = [T_0 + \theta]\tilde{\eta}_\theta(S_A, \theta)$ is obeyed, just as using

the Gibbs function is much preferred to using the two functions $h(S_A, T, P)$ and $\eta(S_A, T, P)$ because the consistency requirement $h_T = T\eta_T$ is automatically satisfied when using the Gibbs function. By contrast, the use of one of the thermodynamic potentials of this paper in terms of $(S_A, \Theta, P)$ is not required to ensure any consistency property. Rather the existence of these thermodynamic potentials, (13 or 14), (A1) and (A2) provides a thermodynamic "completeness" to the use of Conservative Temperature in marine science. The use of one of these thermodynamic potentials, (13 or 14), (A1) and (A2) is equivalent to simply using the combination of $\hat{h}(S_A, \Theta, P)$ and $\hat{\eta}(S_A, \Theta)$.

## Appendix B: The polynomial-based expression for entropy

The polynomial-based expression for specific entropy as a function of Absolute Salinity and Conservative Temperature is given by Eqn. (21) as the sum of the two dominant logarithm terms plus an eighth-order polynomial in the two dimensionless variables $s = [S_A/S_{SO}]^{0.5}$ and $\tau = \Theta/40°C$, where $S_{SO} = 35.165\,04\text{ g kg}^{-1}$ is the Standard Ocean Reference Salinity (IOC *et al.* (2010)),

$$\hat{\eta}(S_A, \Theta) = c_p^0 \ln(1 + \Theta/T_0) + a(S_A/S_{SO}) \ln(S_A/S_{SO}) + P\{8,8\}(s, \tau), \tag{B1}$$

where $T_0 = 273.15\text{K}$ is the Celsius zero point, $c_p^0 = 3991.867\,957\,119\,63\text{ J kg}^{-1}\text{ K}^{-1}$, and the least-squares fit gives the constant $a = -9.309\,495\,003\,228\,781\text{ J kg}^{-1}\text{ K}^{-1}$ and the eighth order polynomial coefficients given by

$P\{8,8\}(s, \tau) =$
$(((((((ETA08*\tau+ETA17*s+ETA07)*\tau$
$+ (ETA26*s+ETA16)*s+ETA06)*\tau$
$+ ((ETA35*s+ETA25)*s+ETA15)*s+ETA05)*\tau$
$+ (((ETA44*s+ETA34)*s+ETA24)*s+ETA14)*s+ETA04)*\tau \tag{B2}$
$+ ((((ETA53*s+ETA43)*s+ETA33)*s+ETA23)*s+ETA13)*s+ETA03)*\tau$
$+ (((((ETA62*s+ETA52)*s+ETA42)*s+ETA32)*s+ETA22)*s+ETA12)*s+ETA02)*\tau$
$+ ((((((ETA71*s+ETA61)*s+ETA51)*s+ETA41)*s+ETA31)*s+ETA21)*s+ETA11)*s+ETA01)*\tau$
$+ (((((((ETA80*s+ETA70)*s+ETA60)*s+ETA50)*s+ETA40)*s+ETA30)*s+ETA20)*s+ETA10)*s+ETA00$

and the 45 constants (each of which has units of $\text{J kg}^{-1}\text{ K}^{-1}$) are given by

ETA00 = -3.7102436569e-01; ETA10 = 3.0834502223e-04; ETA20 = -3.2916987818e+00;
ETA30 = 7.2818259040e+00; ETA40 = -5.6657256773e+00; ETA50 = 2.8402903938e+00;
ETA60 = -8.9615123138e-01; ETA70 = 1.0035964794e-01; ETA80 = 1.8140964105e-03;
ETA01 = 3.0779211774e-02; ETA11 = 1.5006196848e-03; ETA21 = 1.2029316021e-01;
ETA31 = 3.7464975805e-01; ETA41 = -6.0590428227e-01; ETA51 = 6.4365865093e-02;
ETA61 = 2.4626795446e-02; ETA71 = -1.0335853091e-02; ETA02 = 2.3045093877e+00;
ETA12 = -5.4154968624e-03; ETA22 = -2.5098282844e+00; ETA32 = 1.9163697628e-02;
ETA42 = 9.6230320461e-02; ETA52 = 3.7953034101e-02; ETA62 = -5.1206778774e-04;
ETA03 = -8.4974032876e-01; ETA13 = -1.3727475447e-02; ETA23 = 8.6969911602e-01;
ETA33 = 1.1127539375e-01; ETA43 = -8.7616123860e-02; ETA53 = -1.6250024449e-02;
ETA04 = 4.1807750439e-01; ETA14 = 5.1388181100e-02; ETA24 = -3.1917000611e-01;
ETA34 = -4.4999965986e-02; ETA44 = 3.3822211876e-02; ETA05 = -1.9191736060e-01;
ETA15 = -5.3890029514e-02; ETA25 = 9.3472917957e-02; ETA35 = -4.9779616704e-04;
ETA06 = 6.6066546976e-02; ETA16 = 2.4144978278e-02; ETA26 = -1.2850921670e-02;
ETA07 = -1.3678360946e-02; ETA17 = -4.1337102429e-03; ETA08 = 1.1180283076e-03;

**Appendix C: Expressions for thermodynamic variables in terms of $\hat{h}(S_A, \Theta, P)$, $\hat{\eta}(S_A, \Theta)$ and $\hat{\phi}(S_A, \Theta, P)$**

**C.1 Expressions for entropy and enthalpy in terms of $g(S_A, T, P)$ and $\widehat{\phi}(S_A, \Theta, P)$**

Eqns. (15) and (16) for entropy $\eta$ and enthalpy $h$ in terms of $\hat{\phi}(S_A, \Theta, P)$ are compared to the corresponding expressions for these variables in terms of the Gibbs function $g(S_A, T, P)$,

$$\eta = -\hat{\phi}_\Theta(S_A, \Theta, P_0) = -\hat{\phi}_\Theta(S_A, \Theta, P) + \int_{P_0}^{P} \hat{\phi}_{P\Theta}(S_A, \Theta, P')\mathrm{d}P'$$

$$= -g_T(S_A, T, P) = -g_T(S_A, T, P_0) - \int_{P_0}^{P} g_{PT}(S_A, T, P')\mathrm{d}P'. \tag{C1}$$

and

$$h = c_p^0\Theta + \hat{\phi}(S_A, \Theta, P) - \hat{\phi}(S_A, \Theta, P_0) = c_p^0\Theta + \int_{P_0}^{P} \hat{\phi}_P(S_A, \Theta, P')\mathrm{d}P'$$

$$= g(S_A, T, P) - Tg_T(S_A, T, P) = h(S_A, T, P_0) + \int_{P_0}^{P} g_P(S_A, T, P')\mathrm{d}P' - T\int_{P_0}^{P} g_{PT}(S_A, T, P')\mathrm{d}P'. \tag{C2}$$

There are some similarities between these expressions using the two different thermodynamic potentials, and there are differences. When expressed using Conservative Temperature, $\hat{\eta}(S_A, \Theta)$ is not a separate function of pressure, so that in the first line of Eqn. (C1), where $-\hat{\phi}_\Theta(S_A, \Theta, P)$ is evaluated at pressure $P$, this pressure dependence needs to be subtracted. In Eqn. (C2) note that $h(S_A, T, P_0)$ is not the same as potential enthalpy $c_p^0\Theta$ except when the in-situ pressure $P$ happens to be $P_0$.

**C.2 Variables expressed using $\hat{h}(S_A, \Theta, P)$ and $\hat{\eta}(S_A, \Theta)$ compared with $h(S_A, T, P)$ and $\eta(S_A, T, P)$**

Considering changes occurring at constant Absolute Salinity and pressure, the FTR in the forms Eqns. (8) and (11) show that in situ temperature $T = T_0 + t$ is given by

$$T = \partial h/\partial\eta|_{S_A, P} = h_T/\eta_T = \hat{h}_\Theta/\hat{\eta}_\Theta. \tag{C3}$$

The $h_T/\eta_T$ part of this equation is a consistency requirement between the temperature dependence of the $h(S_A, T, P)$ and $\eta(S_A, T, P)$ expressions. That is, expressions for $h(S_A, T, P)$ and $\eta(S_A, T, P)$ cannot be formed independently of each other but rather must satisfy the consistency relationship, $T = h_T/\eta_T$, since $T$ is one of the independent variables. If necessary, however, the required consistency may be established by the integration,

$$\eta(S_A, T, P) = \int_{T_0}^{T} \frac{h_T(S_A, T', P)}{T'} \, dT' + \eta(S_A, T_0, P), \tag{C4}$$

so that $h(S_A, T, P)$ in combination with an independent function $\eta(S_A, T_0, P)$ taken at an arbitrary reference temperature $T_0$ together provide the necessary information. The corresponding relationship in the $(S_A, \Theta, P)$ case, $T = \hat{h}_\Theta/\hat{\eta}_\Theta$, does not impose any such consistency requirement on $\hat{h}(S_A, \Theta, P)$ or $\hat{\eta}(S_A, \Theta)$ because $T$ is not an independent variable in this case.

The expression for specific volume in terms of the Gibbs function is very neat and compact, being $v = g_P$, while the corresponding expression in terms of $h(S_A, T, P)$ and $\eta(S_A, T, P)$ is $v = h_P - (h_T/\eta_T)\eta_P$ (see Eqn. 8). Since $S_A$, $\Theta$ and $\eta$ are all "potential" variables, when the material derivative of enthalpy in the FTR is expressed in the form $\hat{h}_{S_A}\mathrm{d}S_A + \hat{h}_\Theta\mathrm{d}\Theta + \hat{h}_P\mathrm{d}P$,

one finds (from Eqn. (11) by considering the adiabatic and isohaline situation when $dS_A = d\eta = d\Theta = 0$) that specific volume is given by $\hat{h}_P$, hence we have

$$v = g_P = h_P - (h_T/\eta_T)\eta_P = \hat{h}_P. \tag{C5}$$

Note that specific volume can also be expressed in terms of $\hat{h}(S_A, \Theta, P)$ and $\hat{\eta}(S_A, \Theta)$ as $v = \hat{h}_P - (\hat{h}_\Theta/\hat{\eta}_\Theta)\hat{\eta}_P$ because $\hat{\eta}_P$ is zero, and so the last two equalities in Eqn. (C5) are more similar than they appear to be.

In terms of the Gibbs function, the adiabatic lapse rate (the rate of change of in situ temperature during an adiabatic and isohaline change in pressure, see McDougall and Feistel, 2003), is $\Gamma = -g_{TP}/g_{TT}$, while using the two expressions in terms of enthalpy and entropy gives (by differentiating $\hat{h}_\Theta/\hat{\eta}_\Theta$ (from Eqn. C3) with respect to pressure)

$$\Gamma = -g_{TP}/g_{TT} = -\eta_P/\eta_T = \hat{h}_{P\Theta}/\hat{\eta}_\Theta = v_T/\eta_T = \hat{v}_\Theta/\hat{\eta}_\Theta, \tag{C6}$$

where the last two expressions are written in terms of specific volume and entropy. Another expression for $\Gamma$ that corresponds to $-\eta_P/\eta_T$ is $-(\partial\Theta/\partial P|_{S_A,T})/(\partial\Theta/\partial T|_{S_A,P})$.

The relative chemical potential $\mu = g_{S_A} = \partial h/\partial S_A|_{\eta,P} = \tilde{\tilde{h}}_{S_A}$ can be expressed as (from the partial differentials in Eqn. (8) and (11))

$$\mu = h_{S_A} - (h_T/\eta_T)\eta_{S_A} = \hat{h}_{S_A} - (\hat{h}_\Theta/\hat{\eta}_\Theta)\hat{\eta}_{S_A}, \tag{C7}$$

and the chemical potential of water in seawater, $\mu^W = g - S_A g_{S_A}$, can be expressed as

$$\mu^W = (h - S_A h_{S_A}) - (h_T/\eta_T)(\eta - S_A\eta_{S_A}) = (\hat{h} - S_A\hat{h}_{S_A}) - (\hat{h}_\Theta/\hat{\eta}_\Theta)(\hat{\eta} - S_A\hat{\eta}_{S_A}). \tag{C8}$$

Again, it is interesting that these expressions for $\mu$ and $\mu^W$, written in terms of enthalpy and entropy, have the same form whether as functions of $(S_A, T, P)$ or $(S_A, \Theta, P)$.

The adiabatic and isohaline compressibility has the following compact expression in terms of $\hat{h}(S_A, \Theta, P)$

$$\kappa = -\hat{h}_{PP}/\hat{h}_P, \tag{C9}$$

but because in situ temperature does not possess the "potential" property the expressions in terms of $(S_A, T, P)$ are not as compact, being,

$$\kappa = -g_{PP}/g_P + (g_{TP})^2/(g_P g_{TT}) = -(h_{PP}\eta_T - h_T\eta_{PP} + \eta_P^2)/(h_P\eta_T - h_T\eta_P). \tag{C10}$$

It is interesting that $\kappa$ can also be expressed by the same expression as this last one in Eqn. (C10) even when enthalpy and entropy are functions of $(S_A, \Theta, P)$, namely as $-(\hat{h}_{PP}\hat{\eta}_\Theta - \hat{h}_\Theta\hat{\eta}_{PP} + \hat{\eta}_P^2)/(\hat{h}_P\hat{\eta}_\Theta - \hat{h}_\Theta\hat{\eta}_P)$, because $\hat{\eta}_P$ and $\hat{\eta}_{PP}$ are both zero. That is, the last expressions in Eqns. (C9) and (C10) are more similar than they appear to be.

These expressions for the various thermodynamic variables are summarized in Table 2.

## C.3 The constraints on thermodynamic variables revealed by cross-differentiation

When we take the second order cross derivatives of the thermodynamic potential $\tilde{\tilde{h}}(S_A, \eta, P)$, we find the following relations between the observed quantities $v, T$ and $\mu$,

$$\widetilde{T}_P = \widetilde{v}_\eta, \tag{C11}$$

$$\widetilde{\mu}_P = \widetilde{v}_{S_A}, \tag{C12}$$

$$\widetilde{\mu}_\eta = \widetilde{T}_{S_A}, \tag{C13}$$

and the second order cross derivatives of the Gibbs function $g(S_A, T, P)$ give the following relations between the observed quantities $v(S_A, T, P)$, $\eta(S_A, T, P)$, and $\mu(S_A, T, P)$ (the so-called Maxwell relationships)

$$-\eta_P = v_T, \tag{C14}$$

$$\mu_P = v_{S_A}, \tag{C15}$$

$$\mu_T = -\eta_{S_A}. \tag{C16}$$

For our new thermodynamic potential, $\hat{\phi}(S_A, \Theta, P)$, we write the total differential of $\hat{\phi}(S_A, \Theta, P)$ in the form (using Eqns. (11) and (14))

$$d\hat{\phi} = \left\{ \hat{\mu} + T\hat{\eta}_{S_A} - \int_0^\Theta \hat{\eta}_{S_A}(S_A, \Theta')d\Theta' \right\} dS_A + \left\{ T\hat{\eta}_\Theta - c_p^0 - \hat{\eta} \right\} d\Theta + \hat{v}dP \tag{C17}$$

 which involves the three partial derivatives,

$$\hat{\phi}_{S_A} = \hat{\mu} + T\hat{\eta}_{S_A} - \int_0^\Theta \hat{\eta}_{S_A}(S_A, \Theta')d\Theta', \tag{C18}$$

$$\hat{\phi}_\Theta = T\hat{\eta}_\Theta - c_p^0 - \hat{\eta}, \tag{C19}$$

$$\hat{\phi}_P = \hat{v}, \tag{C20}$$

so that the three cross-derivatives yield

$$\hat{T}_P \hat{\eta}_\Theta = \hat{v}_\Theta, \tag{C21}$$

$$\hat{\mu}_P + \hat{T}_P \hat{\eta}_{S_A} = \hat{v}_{S_A}, \tag{C22}$$

$$\hat{\mu}_\Theta + \hat{T}_\Theta \hat{\eta}_{S_A} = \hat{T}_{S_A} \hat{\eta}_\Theta, \tag{C23}$$

after subtracting the two terms $-\hat{\eta}_{S_A}$ and $\hat{T}\hat{\eta}_{S_A\Theta}$ that appear in both $\hat{\phi}_{S_A\Theta}$ and $\hat{\phi}_{\Theta S_A}$ and would have appeared on both sides of Eqn. (C23).

 Note that the equality between $-\eta_P$ and $v_T$ in (C14) does not resemble the balance $\hat{T}_P \hat{\eta}_\Theta = \hat{v}_\Theta$ in (C21), and moreover we know that the corresponding pressure derivative of entropy, $\hat{\eta}_P$, is zero. Rather, the expression (C21) for the adiabatic lapse rate, $\Gamma = \widetilde{T}_P = \hat{T}_P = (\hat{v}_\Theta / \hat{\eta}_\Theta)$, resonates with the result $\widetilde{T}_P = \widetilde{v}_\eta$ of Eqn. (C11). The additional term $\hat{T}_P \hat{\eta}_{S_A} = (\hat{v}_\Theta / \hat{\eta}_\Theta)\hat{\eta}_{S_A}$ in Eqn. (C22) compared with the corresponding formulae in Eqns. (C12) or (C15) is small (being less than 0.5% of both $\hat{v}_{S_A}$ and $\hat{\mu}_P$). The relationship (C23) that comes from equating $\hat{\phi}_{S_A\Theta}$ and $\hat{\phi}_{\Theta S_A}$ has some similarities with both (C13) and (C16),
 with $\hat{T}_\Theta \hat{\eta}_{S_A}$ appearing to be an additional term in one case and $\hat{T}_{S_A} \hat{\eta}_\Theta$ in the other case.

It can be shown by coordinate transformation that each of (C21) – (C23) contains exactly the same information as do (C14) – (C16). This is, each of the equations (C21) – (C23) can be found by transforming the corresponding equation in (C14) – (C16) from $(S_A, T, P)$ coordinates into $(S_A, \Theta, P)$ coordinates.

**Appendix D: Deducing the FTR from the differential of a thermodynamic potential and its definition in terms of**

**enthalpy and entropy.**

The Fundamental Thermodynamic Relationship (FTR) can be deduced from knowledge of the total differential of the Gibbs function $dg = \mu dS_A - \eta dT + v dP$ together the definition of the Gibbs function in terms of enthalpy and entropy, $g \equiv h - T\eta$. Here we demonstrate the corresponding result for $\hat{\phi}(S_A, \Theta, P)$, namely that the FTR can be found from knowledge of the

total differential of $\hat{\phi}(S_A, \Theta, P)$ as well as its definition in terms of enthalpy and entropy.

We write the total differential of $\hat{\phi}(S_A, \Theta, P)$ in the form of Eqn. (C17)

$$d\phi = \left\{ \hat{\mu} + T\hat{\eta}_{S_A} - \int_0^\Theta \hat{\eta}_{S_A}(S_A, \Theta') d\Theta' \right\} dS_A + \left\{ T\hat{\eta}_\Theta - c_p^0 - \hat{\eta} \right\} d\Theta + \hat{v}(S_A, \Theta, P) dP \tag{D1}$$

and we use the definition of $\hat{\phi}(S_A, \Theta, P)$ in the form Eqn. (14), repeated here,

$$\hat{\phi}(S_A, \Theta, P) = \hat{h}(S_A, \Theta, P) - \int_0^\Theta [c_p^0 + \hat{\eta}(S_A, \Theta')] d\Theta', \tag{D2}$$

and we ask whether the FTR can be deduced from knowledge of Eqns. (D1) and (D2), in direct analogy to what is possible for the Gibbs function.

Because of the definition of Conservative Temperature, $c_p^0 \Theta \equiv \hat{h}(S_A, \Theta, P_0)$, we know that $\eta = \hat{\eta}(S_A, \Theta)$, $\hat{h}_\Theta(S_A, \Theta, P_0) = c_p^0$ and $\hat{h}_{S_A}(S_A, \Theta, P_0) = 0$. Equating the three partial derivatives that appear in Eqn. (D1) with the corresponding expressions from differentiating Eqn. (D2) shows that $\hat{h}_{S_A} = \mu + T\hat{\eta}_{S_A}$, $\hat{h}_\Theta = T\hat{\eta}_\Theta$ and $\hat{h}_P = v$, so that the expression, Eqn. (11), for the

total derivative of enthalpy has been found. Using $d\eta = \hat{\eta}_{S_A} dS_A + \hat{\eta}_\Theta d\Theta$, the FTR, Eqn. (2), follows, and the analogy with the Gibbs function is complete.

**Code availability**

Upon acceptance of this paper for publication, the 24 Gibbs SeaWater Oceanographic Toolbox (GSW) subroutines in Matlab that we have prepared will replace existing subroutines of the same name that are presently in the GSW code on the TEOS-10 web site ( http://www.teos-10.org/software.htm ).

**Author contribution**

TMcD discovered the forms, Eqns. (14), (A1)-(A3), (A8) and (A11), of the thermodynamic potential of seawater, PMB wrote and tested the 24 new computer subroutines that were needed to implement the ideas of this paper in the GSW computer software of TEOS-10, RF ensured that the thermodynamic reasoning in the paper was precise, while FR performed the fit of entropy to Absolute Salinity and Conservative Temperature. All authors contributed to writing the manuscript.

**Acknowledgements**

We thank William Dewar, Stephen Griffies and Remi Tailleux for many helpful suggestions which improved the paper. TMcD and PMB gratefully acknowledge Australian Research Council support through grant FL150100090. This paper contributes to the tasks of the IAPSO/SCOR/IAPWS Joint Committee on the Properties of Seawater.

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

**Figure Captions**

**Figure 1.** Panels (a) and (b) are contour plots of $c_p^0 \ln(1 + \theta/T_0) - \eta$ and $c_p^0 \ln(1 + \Theta/T_0) - \eta$ respectively, while panels

(c) and (d) show $c_p^0 \ln(1 + \theta/T_0) + a(S_A/S_{SO}) \ln(S_A/S_{SO}) - \eta$ and $c_p^0 \ln(1 + \Theta/T_0) + a(S_A/S_{SO}) \ln(S_A/S_{SO}) - \eta$

respectively. All panels in this figure are in the units of entropy, namely J kg$^{-1}$ K$^{-1}$.

**Figure 2.** (a) The error in the fit Eqn. (21) to entropy (in units of $10^{-6}$ J kg$^{-1}$ K$^{-1}$). (b) The error in evaluating potential

temperature $\theta$ (in $\mu K$) from Eqns. (21) and (22).

**Table 1: Table of symbols.**

| Quantity | Symbol | Units | Comments |
|---|---|---|---|
| Standard Ocean Reference Salinity | $S_{\text{SO}}$ | g kg$^{-1}$ | 35.165 04 g kg$^{-1}$, corresponding to the standard ocean Practical Salinity of 35. |
| Absolute Pressure | $P$ | Pa | When Absolute Pressure is used it should always be in Pa, not in Mpa, nor in dbar. |
| sea pressure. Sea pressure is the pressure argument to all the GSW Toolbox functions. | $p$ | dbar | Equal to $P - P_0$ and usually expressed in dbar not Pa. |

| Quantity | Symbol | Units | Comments |
|---|---|---|---|
| one standard atmosphere | $P_0$ | Pa | exactly 101 325 Pa ($= 10.1325$ dbar) |
| Absolute Salinity | $S_{\text{A}} = S_{\text{A}}^{\text{dens}}$ | g kg$^{-1}$ | Absolute Salinity is measured on the Millero *et al.* (2008) Reference-Salinity Scale. |

| Quantity | Symbol | Units | Comments |
|---|---|---|---|
| temperature | $t$ | ºC | |
| Absolute Temperature | $T$ | K | $T/K \equiv T_0/K + t/(\text{ºC}) = 273.15 + t/(\text{ºC})$ |
| temperature derivatives | $T$ | K | When a quantity is differentiated with respect to *in situ* temperature, the symbol $T$ is used in order to distinguish this variable from time. |
| Celsius zero point | $T_0$ | K | $T_0 \equiv 273.15$ K |
| potential temperature | $\theta$ | ºC | |
| Conservative Temperature | $\Theta$ | ºC | |
| the "specific heat", for use with Conservative Temperature | $c_p^0$ | J kg$^{-1}$ K$^{-1}$ | $c_p^0 \equiv 3991.867\,957\,119\,63$ J kg$^{-1}$ K$^{-1}$  This 15-digit number is defined to be the exact value of $c_p^0$. $c_p^0$ is the ratio of potential enthalpy $h^0$ to $\Theta$. |
| specific enthalpy | $h$ | J kg$^{-1}$ | $h = u + Pv$ |
| specific potential enthalpy | $h^0$ | J kg$^{-1}$ | specific enthalpy referenced to zero sea pressure. |
| specific isobaric heat capacity | $c_p$ | J kg$^{-1}$ K$^{-1}$ | $c_p = \partial h/\partial T|_{S_{\text{A}},P}$ |
| specific internal energy | $u$ | J kg$^{-1}$ | |

| | | | |
|---|---|---|---|
| specific Gibbs function (Gibbs energy) | $g$ | J kg$^{-1}$ | |
| specific Helmholtz energy | $f$ | J kg$^{-1}$ | |
| specific entropy | $\eta$ | J kg$^{-1}$ K$^{-1}$ | |
| density | $\rho$ | kg m$^{-3}$ | |

| | | | |
|---|---|---|---|
| thermal expansion coefficient with respect to Conservative Temperature $\Theta$ | $\alpha^{\Theta}$ | K$^{-1}$ | $\alpha^{\Theta} = v^{-1}\partial v/\partial\Theta\vert_{S_A,P} = -\rho^{-1}\partial\rho/\partial\Theta\vert_{S_A,P}$ |
| saline contraction coefficient at constant Conservative Temperature | $\beta^{\Theta}$ | kg g$^{-1}$ | $\beta^{\Theta} = -v^{-1}\partial v/\partial S_A\vert_{\Theta,P} = \rho^{-1}\partial\varrho/\partial S_A\vert_{\Theta,P}$<br>Note that the units for $\beta^{\Theta}$ in the GSW Oceanographic Toolbox are consistent with $S_A$ being in g kg$^{-1}$. |
| isothermal compressibility | $\kappa^t$ | Pa$^{-1}$ | |
| isentropic and isohaline compressibility | $\kappa$ | Pa$^{-1}$ | |
| chemical potential of water in seawater | $\mu^W$ | J g$^{-1}$ | |
| chemical potential of sea salt in seawater | $\mu^S$ | J g$^{-1}$ | |
| relative chemical potential of (sea salt and water in) seawater | $\mu$ | J g$^{-1}$ | $\mu = (\partial g/\partial S_A)_{T,P} = (\partial h/\partial S_A)_{\eta,P} = \mu^S - \mu^W$ |
| dissipation rate of kinetic energy per unit mass | $\varepsilon$ | J kg$^{-1}$ s$^{-1}$<br>$= $ m$^2$ s$^{-3}$ | |
| adiabatic lapse rate | $\Gamma$ | K Pa$^{-1}$ | |
| sound speed | $c$ | m s$^{-1}$ | |
| specific volume | $v$ | m$^3$ kg$^{-1}$ | $v = \rho^{-1}$ |

**Table 2. Expressions for various thermodynamic variables based on different thermodynamic potentials**

| | Expressions based on $\widetilde{h}(S_A, \eta, P)$ | Expressions based on $g(S_A, T, P)$ | Expressions based on $h(S_A, T, P)$ and $\eta(S_A, T, P)$ | Expressions based on $\hat{h}(S_A, \Theta, P)$ and $\hat{\eta}(S_A, \Theta)$ |
|---|---|---|---|---|
| $T$ | $T = \widetilde{h}_\eta$ | $T$ | $T = h_T/\eta_T$ <br> This is a necessary consistency condition between $h(S_A, T, P)$ and $\eta(S_A, T, P)$. | $T = (T_0 + t) = \hat{h}_\Theta/\hat{\eta}_\Theta$ |
| $\theta$ | $T_0 + \theta = \widetilde{h}_\eta(S_A, \eta, P_0)$ | $g_T(S_A, T_0 + \theta, P_0) = g_T(S_A, T, P)$ <br> This is an implicit equation for $\theta$. | $\eta(S_A, T_0 + \theta, P_0) = \eta(S_A, T, P)$ <br> This is an implicit equation for $\theta$. | $(T_0 + \theta) = c_p^0/\hat{\eta}_\Theta$ |
| $\Theta$ | $\Theta = \widetilde{h}(S_A, \eta, P_0)/c_p^0$ | $\Theta = g(S_A, T_0 + \theta, P_0)/c_p^0$ <br> $- (T_0 + \theta)g_T(S_A, T_0 + \theta, P_0)/c_p^0$ | $\Theta = h(S_A, T_0 + \theta, P_0)/c_p^0$ | $\Theta; \quad \Theta \equiv \hat{h}(S_A, \Theta, P_0)/c_p^0$ |
| $h$ | $\widetilde{h}(S_A, \eta, P)$ | $g - Tg_T$ | $h(S_A, T, P)$ | $\hat{h}(S_A, \Theta, P)$ |
| $g$ | $\widetilde{h} - \eta\widetilde{h}_\eta$ | $g(S_A, T, P)$ | $h - T\eta$ | $\hat{h} - \hat{\eta}\,\hat{h}_\Theta/\hat{\eta}_\Theta$ |
| $\eta$ | $\eta$ | $-g_T$ | $\eta(S_A, T, P)$ | $\hat{\eta}(S_A, \Theta)$ |
| $v$ | $\widetilde{h}_P$ | $g_P$ | $h_P - T\eta_P$ | $\hat{h}_P$ |
| $u$ | $\widetilde{h} - P\widetilde{h}_P$ | $g - Tg_T - Pg_P$ | $h - Ph_P + TP\eta_P$ | $\hat{h} - P\hat{h}_P$ |
| $\mu$ | $\widetilde{h}_{S_A}$ | $g_{S_A}$ | $h_{S_A} - T\eta_{S_A}$ | $\hat{h}_{S_A} - \hat{\eta}_{S_A}\hat{h}_\Theta/\hat{\eta}_\Theta$ |
| $\mu^W$ | $\widetilde{h} - \eta\widetilde{h}_\eta - S_A\widetilde{h}_{S_A}$ | $g - S_A g_{S_A}$ | $\left(h - S_A h_{S_A}\right) - T\left(\eta - S_A\eta_{S_A}\right)$ | $\left(\hat{h} - S_A\hat{h}_{S_A}\right)$ <br> $-\left(\hat{h}_\Theta/\hat{\eta}_\Theta\right)\left(\hat{\eta} - S_A\hat{\eta}_{S_A}\right)$ |
| $f$ | $\widetilde{h} - \eta\widetilde{h}_\eta - P\widetilde{h}_P$ | $g - Pg_P$ | $(h - T\eta) - P(h_P - T\eta_P)$ | $\hat{h} - \hat{\eta}\,\hat{h}_\Theta/\hat{\eta}_\Theta - P\hat{h}_P$ |
| $\kappa$ | $-\widetilde{h}_{PP}/\widetilde{h}_P$ | $-g_{PP}/g_P + (g_{TP})^2/(g_P g_{TT})$ | $-(h_{PP}\eta_T - h_T\eta_{PP} + \eta_P^2)/(h_P\eta_T - h_T\eta_P)$ | $-\hat{h}_{PP}/\hat{h}_P$ |
| $\Gamma$ | $\widetilde{h}_{P\eta}$ | $-g_{TP}/g_{TT}$ | $-\eta_P/\eta_T$ | $\hat{h}_{P\Theta}/\hat{\eta}_\Theta$ |
| $\alpha^\Theta$ | $\dfrac{c_p^0 \widetilde{h}_{P\eta}}{\widetilde{h}_P \widetilde{h}_\eta(S_A, \eta, P_0)}$ | $-\dfrac{g_{PT}}{g_P}\dfrac{c_p^0}{(T_0 + \theta)g_{TT}}$ | $-\dfrac{\eta_P}{(h_P - T\eta_P)}\dfrac{c_p^0}{(T_0 + \theta)\eta_T}$ | $\hat{h}_{P\Theta}/\hat{h}_P$ |
| $\beta^\Theta$ | $-\dfrac{\widetilde{h}_{PS_A}}{\widetilde{h}_P} + \dfrac{\widetilde{h}_{P\eta}}{\widetilde{h}_P}\dfrac{\widetilde{h}_{S_A}(S_A,\eta,P_0)}{\widetilde{h}_\eta(S_A,\eta,P_0)}$ | Expression too large to fit here | Expression too large to fit here | $-\hat{h}_{PS_A}/\hat{h}_P$ |