# Peer review of "A Thermodynamic Potential of Seawater in terms of Absolute Salinity, Conservative Temperature and *in situ* Pressure"

_EGUsphere, 2023_

## Referee Comment (RC2)

Review of

**A Thermodynamic Potential of Seawater**
**in terms of Conservative Temperature**

by McDougall, Barker, Feistel, and Roquet

Ocean Sciences manuscript https://doi.org/10.5194/egusphere-2023-1568

Stephen.Griffies@noaa.gov

August 6, 2023

**1 Recommendation**

This is an impressive work that comes after decades of investigations by the authors into the fundamentals and practices of seawater thermodynamics. I suspect that this paper will be read for decades to come. I fully support publication, and offer comments targeting clarification in hopes that a few more readers will feel inspired to enter into the seawater thermodynamics club, including those readers who are experts in thermodynamics and yet not versed in the ocean notation (e.g., a theoretical physical chemist).

Before starting the formal review, I note that another reviewer pointed out the need to give a name to $\hat{\phi}(S_A, \Theta, P)$. A generic name could be "Seawater Thermodynamic Potential (STP)", which is a somewhat more concise term than the authors' "thermodynamic potential of seawater in terms of Conservative Temperature". However, there are other thermodynamic potentials assigned a name associated with its proponent (e.g., Gibbs, Helmholtz, Massieu). It is with this view in mind that I recommend we refer to $\hat{\phi}(S_A, \Theta, P)$ as the *McDougall thermodynamic function*. That name is certainly not something for the current paper. But in all seriousness I recommend that the community pick up this terminology moving forward.

**2 General comments**

Here are some general comments.

**2.1 Title**

Title: the key novelty in this work is to develop the theory and practice for a thermodynamic potential using $\Theta$ rather than *in situ* temperature. This novelty is reflected in the title. However, as noted in the first sentence of the abstract, what is in fact proposed is a thermodynamic potential that is a function of $S_A, \Theta, p$. I encourage the authors to use the more complete: "A Thermodynamic Potential of Seawater in terms of Absolute Salinity, Conservative Temperature, and *in situ* pressure". I am particularly keep to allow readers who are not oceanographers to find this title offers a better sense for what is proposed.

**2.2 Notation introduced in lines 75-80**

In between lines 75-80 there is a single sentence that introduces notation all within the text (i.e., no equation numbers). For those having read other papers in the TEOS-10 family, the notation will be familiar even if one does not like it. Indeed, I have been an author on such papers. Even so, as a reviewer I put on my non-TEOS-10 aficionado hat in hopes of identifying places where readers can be confused.

1. The symbol "t" is almost universally used for time in physics and chemistry. Indeed, you use it for time in the First Law equation (1). However, on lines 77, 78, 79, $t$ is used for *in situ* temperature measured in Celsius units. Should you wish to retain this non-standard usage, care should be made to reduce confusion when switching between time and temperature, with signposts placed where needed.

2. $T_0$ is a constant offset from $t$, presumably since $t + T_0$ is the absolute thermodynamic temperature with $t$ in Celsius units. That point is never made yet it would serve the reader to make it here.

3. $T$ is a symbol attached as a subscript to the Gibbs potential, and yet it is never defined. Presumably $T = t + T_0$ is the thermodynamic temperature, but the reader should not need to guess.

4. The above incomplete discussion of $t$ and $T_0$ make me wonder if they are even needed at this point of the presentation. Perhaps you prefer to keep them here since you prefer to measure all temperature quantities in Celsius, and so need the $T_0$ offset. But that is a choice that is not fundamental and can lead to rather awkward equations. Indeed, later in the presentation of the same section, you write $\nabla(1/T)$ rather than the more awkward $\nabla(1/(t + T_0))$.

5. It is stated that subscripts denote partial differentiation, whereas we also find many subscripted symbols that are not derivatives. For example, between lines 75-80 we find $P_r$, $S_A$, and $T_0$. I realize the authors are fond of the subscript shorthand for partial derivatives. Even so, I will poke at them by noting that $\partial_T$ and $\partial_\Theta$ and $\partial_p$ require only a bit more ink on the printed page and yet they offer far less room for notational ambiguity.

6. What does "forward expression" mean on line 78? This term is also used later at lines 480, 481, 486, and 523. It is only when reaching lines 486 and 523 that we find out that "forward calculation" means that no iteration is needed. Please define this term at line 78.

7. $h$ is used in two separate equations between lines 75-80, and yet it is not defined until line 90 in a different section.

**2.3 Clarifying "inconsistencies"**

On line 80 it is stated that "While the inconsistencies in temperature are small". Although contained in the build up material prior to lines 75, it should be emphasized that the ambiguity is not related to a problem with seawater thermodynamics. Rather, it is that we can compute $t$ and $\theta$ using either the "forward expression" on line 79 or the implicit Gibbs equation on line 77, and yet these two expressions, in practice, lead to slightly different numerical values. If you clean up lines 75-80, for example by splitting this material into two or three sentences, then the reader will be a bit more clear on the inconsistency.

**2.4 Further citations for First Law**

Line 85-95 presents the First Law of Thermodynamics as applied to moving seawater fluid. Perhaps the most lucid and correct discussion of this equation is given in TEOS-10, with citations given here to the relevant sections. Even so, I recommend including some of the other places that the reader might find it discussed. In particular, it is worth pointing to Chapters 49 and 58 of *Landau and Lifshitz* (1987) so that our physical chemist reader does not presume TEOS-10 is the first occasion where the First Law was properly derived for a moving multi-component fluid.

**2.5 Infer versus deduce**

Line 97 states that Clausius (1876) "deduced" the existence of entropy. I suggest that it is more proper to say that Clausis "inferred" the existence of entropy. It took the later work of Boltzmann to provide a deductive theory for entropy based on mechanistic and probabalistic principles that led to statistical mechanics.

**2.6 Quasi-static versus reversible**

In much of Section 1.2 (Thermodynamic fundamentals), the authors refer to *reversible processes*. I instead recommend they use the slightly more general, and useful, term *quasi-static*, following the usage given in Sections 4.2 and 4.3 of *Callen* (1985), as well as Sections 2.9 and 2.10 of *Reif* (1965). In these books, the authors define a quasi-static process as a process that moves along a locus of equilibrium states, and with a quasi-static process approximated by a realizable physical process that occurs through steps that are each arbitrarily close to thermodynamic equilibrium. By formulating the notion of a quasi-static process, we are afforded a precise definition for intensive properties such as *in situ* temperature, pressure, and chemical potential, whereas such intensive properties are fuzzy concepts for systems out of thermodynamic equilibrium. A reversible process is a quasi-static process that occurs without net entropy change, and yet not all quasi-static processes are reversible since entropy can generally increase in a quasi-static process.

Here are the lines of text that prompted me to offer my recommendation to switch from "reversible" to "quasi-static". In practice the distinction might be small. But in the spirit of making the fewest assumptions necessary, I recommend switching from your assumed reversible processes to the more general quasi-static processes.

- Line 102 it is stated that the FTR applies to *reversible processes*, and yet *Callen* (1985) emphasizes that the FTR holds for the more general quasi-static processes.

- line 145 states "a seawater parcel is heated reversibly", which I assume means that the parcel's entropy increases when the surrounding environment entropy decreases by the exact same amount. This sort of process is less general than a quasi-static heating of a parcel, in which we do not need to assume zero net entropy production.

- line 149: again change "reversible" to "quasi-static".

- Discussion finishing at line 162. If you replace "reversible" with "quasi-static", then this discussion mirrors that in Section 4.2 and 4.3 of *Callen* (1985).

- If I am barking up the wrong tree, and you do need reversible, then such disagreement with *Callen* (1985) and *Reif* (1965) warrants clarification in your manuscript.

**2.7  Gravity with thermodynamics**

When introducing gravity on line 180, you make the distinction between the distribution of *in situ* temperature and potential temperature in equilibrium. Although a bit tangential to the current discussion, it is worth noting that pressure in thermodynamic equilibrium is not uniform when in a gravity field. Instead pressure is hydrostatically balanced. My motivation for recommending this insertion is that we do not commonly find thermodynamics discussed in the presence of gravity, even though all geophysical fluids are in a gravity field. The notable exeptions include Chapter 9 of *Guggenheim* (1967), §25 of *Landau and Lifshitz* (1980), and Section 1.8 of *Kamenkovich* (1977) (I thank Rainer Feistel for pointing these resources to me in email correspondence in 2022).

**2.8  Criterion of independence**

On line 228 you state that $h(S_A, T, P)$ and $\eta(S_A, T, P)$ violate the criterion of independence given that they must satisfy equation (10). You then say that for the Gibbs potential equation (10) is satisfied. I missed how you know that it is not satisfied for $h(S_A, T, P)$ and $\eta(S_A, T, P)$. One or two more sentences might be sufficient.

Also, equations (9) and (10) have $S, P$ subscripts for the partial derivatives. Should that instead read $S_A, P$?

**2.9  Please build a table of symbols!**

There are many symbols that are introduced in the text and in equations. These symbols are easily forgotten, in which case one needs to sift through the text to find them defined. It would greatly help the reader to have a thorough table of symbols for more easy reference.

**3  Minor comments**

- line 154: I recommend the sentence starting "In practice" should start the beginning of a new paragraph.

- lines 176-177 you state "The hat over a variable indicates that it is being expressed as a function of Conservative Temperature (rather than in situ temperature)." This information would be very useful if stated back around line 75-80 where these symbols are first used.

- line 184 you say "here the cup over a variable's name indicates that it is being expressed as a function of entropy". However, in equation (6) you use the same cup symbol to write entropy as a function of $S_A, h, P$. It would be useful to define this notation back at the point of equation (6).

- Equation (8) has a period whereas there should be none.

- Please see the comments from the other reviewer about lines 253 and 256, where $S$ is written when it should be $S_A$.

- Line 365 there is $\hat{\eta}(S_A, \Theta, P)$ and $\tilde{\eta}(S_A, \Theta, P)$. I do not know what $\tilde{\eta}(S_A, \Theta, P)$ is. Perhaps it is a typo, or perhaps in my wished-for table of math symbols I could find it defined.

- line 353 starts material that, in my opinion, is better placed in Section 4 when detailing how you compute $\hat{\phi}(S_A, \Theta, P)$ in practice.

- line 366: The sentence starting with "In summary" is a great start to a new paragraph.

- line 493: the authors make use of a semi-colon here, whereas a colon is more appropriate. I believe there are a few other occasions of this sort, but I did not mark them on my draft.

**References**

Callen, H. B., *Thermodynamics and an Introduction to Thermostatics*, John Wiley and Sons, New York, 493 + xvi pp, 1985.

Guggenheim, E. A., *Thermodynamics: An Advanced Treatment for Chemists and Physicists*, 5th ed., North-Holland, Amsterdam, 390, 1967.

Kamenkovich, V., *Fundamentals of Ocean Dynamics*, Elsevier Scientific Publishin Company, Amsterdam, 249 pp, 1977.

Landau, L. D., and E. M. Lifshitz, *Statistical Physics: Part 1*, Pergamon Press, Oxford, UK, 544 pp, 1980.

Landau, L. D., and E. M. Lifshitz, *Fluid Mechanics*, Pergamon Press, Oxford, UK, 539 pp, 1987.

Reif, F., *Fundamentals of Statistical and Thermal Physics*, McGraw-Hill, New York, 1965.

---

## Community Comment (CC2)

Comments on: "A thermodynamic potential of seawater in terms of Conservative Temperature" by McDougall et al.

This is an interesting and useful paper, which I enjoyed reading. I have a few remarks about it though, that the authors may want to address.

1. The paper needs a more scholarly review of the theory of thermodynamic potentials. A very good and lucid reference is Alberty (2001). Such a theory highlights at least three key features that the authors appear to have overlooked. The first one is related to the concept of canonical variables. The second one is the theory of Legendre transforms. The third one is the concept of conjugate thermodynamic variables. The first feature is crucial to mention, because a thermodynamic potential contains all possible thermodynamic function only if expressed in terms of canonical variables. The authors should stress the fact that Conservative Temperature (like potential temperature) is not a canonical variable, which is why two functions ($h(\Theta, S, p)$ and $\eta(\Theta, S)$) are needed in that case to predict all possible thermodynamic information about the system. The second feature is crucial to mention, because it is the theory of Legendre transform, and the result that a curve can equivalently be described as the envelope of its tangent lines, which serves to establish the equivalence between the different classical thermodynamic potentials. The third feature is crucial to mention to explain why different thermodynamic potentials have different canonical variables. The author may want to mention that introducing Conservative Temperature (or potential temperature) introduces an external environmental parameter into the system (the reference pressure), which augment the phase space from 3 to 4 dimensions.

2. *Line 213-214: Note that the FTR follows from this expression for the total derivative of the Gibbs function if and only if one also knows that $g = h - T\eta$; we will return to this later.* I don't understand this sentence, because in the theory of thermodynamic potentials, the relation $g = h - T\eta$ is not a matter of knowledge but of definition, in the sense that the relation defines g as the Legendre transform of h. I don't understand what the authors mean by 'if and only if one also knows that [...]'. Do the authors mean: `if one does not know about the theory of thermodynamic potentials and of Legendre transforms'? That seems odd for a paper about thermodynamic potentials.

3. I disagree that the new thermodynamic potential is equivalent to the Gibbs function. Indeed, while it is true that the Gibbs function can be recovered from their newly introduced potential via unambiguous mathematical operations (obtain enthalpy and entropy from their new thermodynamic potential; use the result to eliminate conservative temperature and express specific enthalpy in terms of canonical variables; use the Legendre transform to obtain the Gibbs function), it is not possible to recover the newly defined thermodynamic potential from the Gibbs function without introducing external rules along the way. Indeed, since Conservative Temperature is not conjugate to any canonical variable, human intervention is needed to introduce it by specifying the functional relationship linking it to specific entropy and salinity. Moreover, as the authors demonstrate, there is no unique way to construct a thermodynamic potential from $h(S, \Theta, p)$ and $\eta(S, \Theta)$, so that human

intervention (and ingenuity) is required for that last step. Demonstrating the equivalence between two quantities requires discussion of the steps required to obtain one from the other and conversely.

4.  While I think that explaining how to construct a thermodynamic potential containing all the thermodynamic information when expressed in terms of a non-canonical variable serves a useful purpose, as it clarifies an issue that others may wonder about, I am sceptical that this is of any practical use. As the authors correctly recognise, all necessary information is contained in $h(S, \Theta, p)$ and $\eta(S, \Theta)$, which are independent of each other. It seems pointless (not to say computationally inefficient) to me to construct a thermodynamic potential just as to be able to say that these two functions can be obtained from it. Indeed, the authors make it clear that the thermodynamic potential that they construct is a purely ad-hoc and arbitrary concept with no particular significance. Therefore, while I agree that it is useful to know that such a thermodynamic potential can be constructed in principle, I disagree that it is useful to construct it in practice. To me, it seems more computationally efficient to construct enthalpy and entropy as function of S, CT, and p, and derive all thermodynamic quantities from these two functions without introducing the thermodynamic potential. For this reason, I think that the authors should reconsider the way that they present their material. I think that the correct conclusion that derives from their results is: Yes, it is possible to construct a thermodynamic potential in terms of S, CT, and p, and here is how to do it, which is of theoretical interest for the theory of thermodynamic potentials formulated in terms of non-canonical variables, but this does not appear to offer any practical advantage.

5.  *Line 535. This provides a theoretical boost to using Conservative Temperature as the temperature variable in physical oceanography as recommended by TEOS-10 (Valladares et al., 2011a,b).*
    I don't understand why for at least two reasons. The first one is that as mentioned before, the authors' thermodynamic potential is clearly an artificial and arbitrary object. As a result, it seems to me that associating Conservative Temperature to an artificial object will just end up highlighting the artificial character of Conservative Temperature, the opposite of a theoretical boost. The second one is that the whole machinery described could obviously be equally applied to potential temperature or any other non-canonical variable. In other words, their results are not specific to Conservative Temperature, so it is unclear why they should favour it more than any other non-canonical variable.

```
@article{Alberty1994,
        author = {R. A. Alberty},
        journal = {Chemical Reviews},
        number = {6},
        pages = {1457--1482},
        title = {Legendre transforms in chemical thermodynamics},
        volume = {94},
        year = {1994}}
```

---

## Author Comment (AC1)

**Final Author Comment**

**on the three referees' reports of the manuscript**

**A Thermodynamic Potential of Seawater in terms of Conservative Temperature by McDougall, Barker, Feistel, and Roquet**

The referee's comments are reproduced below in full in black text, and our replies are in red text.

**Referee #1**

The authors describe the derivation of a new thermodynamic potential tailored to oceanography, in that its native variables are absolute salinity, conservative temperature and pressure. They also detail its analysis, demonstrating its equivalency to the sea water Gibbs function, currently the backbone of the TEOS-10 thermodynamic framework in broad oceanographic use. It is argued this potential represents a cleaner foundation for oceanic thermodynamics and offers a (modest) savings in computations, and propose to use the potential in the next generation of TEOS-10 software.

This is quite a paper. At places, it is heavy going, as is often the case with thermodynamics. Having said that, I find the paper pleasantly readable and the main points relatively clear (subject to a few caveats outlined below). While the authors are quick to emphasize that the practical value of the work is somewhat modest, resulting small increases in computational accuracy and savings in computer time, I am impressed by the intellectual achievement, i.e. the discovery of a new thermodynamic potential. And the roadmap by which one gets there, in its description, illustrates a considerable amount of creativity and insight, as well as practical knowledge of the nuts and bolts of operational oceanographic thermodynamics.

**We thank the reviewer for these positive comments.**

It would be nice if the potential were given a name other than 'the Thermodynamic Potential \hat{Phi}'.

**At this stage we refrain from naming the new thermodynamic potential, especially we will refrain from naming it after a living human being.**

Ultimately, I recommend this paper for publication and support the transition to the new potential for use in TEOS-10, although the latter might want to be phased in over some trial period while the field gets some experience with its use. I do have three issues of varying degrees of significance which I now raise which the authors may wish to consider in any revisions. I do not see any of them as insurmountable.

First, around line 255 I believe there are a series of typographical errors mostly involving displaced commas and the dropping of the subscript A on absolute salinity. Although minor, these are the sorts of details that can throw the careful reader for a loss. Thanks; these are now fixed. After that, the authors ask the question in section 3.2 about the equivalency of the new potential to the Gibbs, ultimately answering in the affirmative. The presentation is in its own way convincing, but proceeds by connecting the two potentials in a logical way. More traditionally, the informational equivalence of the various classical potentials is demonstrated by showing they are related to one another via Legendre transforms. Is such a demonstration possible in this case? We now make the point that unlike all known thermodynamic potential prior to our paper, ours are not Legendre transformations of known thermodynamic potentials. We say this now, but we don't know what else to say; ours are just new, and different. Or, perhaps, by Legendre transforming this new potential, other potentials might be uncovered. Last, around line 260, the statement is made that two of the more useful features of potential enthalpy are that its derivative with respect to conservative temperature is heat capacity and its derivative with respect to absolute Salinity vanishes. This proceeds guite simply from the definitions of potential enthalpy and Conservative temperature, provided that one accepts the reduction of entropy from its dependence on three thermodynamic variables to two, due to its 'Potential' property. But, I took it on as an exercise to work my way through the intervening steps, starting with potential enthalpy as a function of salinity and entropy, and introducing a 'potential entropy' variable and formally changing coordinates. I was eventually able to arrive at the conclusion of the authors, but it was a bit of work (assuming I have done everything correctly). Would it be at all useful to include such a demonstration in an appendix? Calculations related to this appear around line 410. This derivation has now been expanded and made clearer, including why entropy is a "potential" function (that is one of its defining characteristics back in the 1880s. In fact, entropy was the original "potential" variable).

Conceptually, I am also interested by the capacity of this new potential to somewhat cleanly separate buoyancy like seawater characteristics from their chemical potential characteristics. I suppose this reflects the very nearly conservative property of Conservative Temperature, but I still suspect there are some profound implications here. I am on the steep part of the learning curve with regards to this paper. The clean separation of some thermodynamic properties depending solely on  $\hat{h}(S_A, \Theta, P)$  is actually due to the "potential" nature of Conservative Temperature, rather than to its nearly conservative nature. While this separation was realized when the TEOS-10 Manual (IOC et al. (2010)) was written in 2019, it is only in the last few years as we have been hatching the ideas in the present paper that we have really appreciated the importance of this clean separation, and indeed the complete independence of the thermodynamic information in  $\hat{h}(S_A, \Theta, P)$  and in  $\hat{\eta}(S_A, \Theta)$ . This does seem to be important, and I'm not sure that we fully understand the importance as yet.

**Referee #2, Stephen Griffies**

Review of

A Thermodynamic Potential of Seawater in terms of Conservative Temperature by McDougall, Barker, Feistel, and Roquet Ocean Sciences manuscript https://doi.org/10.5194/egusphere-2023-1568 Stephen.Griffies@noaa.gov August 6, 2023

**1 Recommendation**

This is an impressive work that comes after decades of investigations by the authors into the fundamentals and practices of seawater thermodynamics. I suspect that this paper will be read for decades to come. I fully support publication, and offer comments targeting clarification in hopes that a few more readers will feel inspired to enter into the seawater thermodynamics club, including those readers who are experts in thermodynamics and yet not versed in the ocean notation (e.g., a theoretical physical chemist).

Before starting the formal review, I note that another reviewer pointed out the need to give a name to  $\hat{\phi}(S_A, \Theta, P)$ . A generic name could be "Seawater Thermodynamic Potential (STP)", which is a somewhat more concise term than the authors' "thermodynamic potential of seawater in terms of Conservative Temperature". However, there are other thermodynamic potentials assigned a name associated with its proponent (e.g., Gibbs, Helmholtz, Massieu). It is with this view in mind that I recommend we refer to  $\hat{\phi}(S_A, \Theta, P)$  as the McDougall thermodynamic function. That name is certainly not something for the current paper. But in all seriousness I recommend that the community pick up this terminology moving forward.

We thank the reviewer for their incredibly detailed review, and for their supportive words. At this stage the lead author has decided to refrain from naming the new thermodynamic potential after a human being.

**2 General comments**

Here are some general comments.

**2.1 Title**

Title: the key novelty in this work is to develop the theory and practice for a thermodynamic potential using  $\Theta$  rather than in situ temperature. This novelty is reflected in the title. However, as noted in the first sentence of the abstract, what is in fact proposed is a thermodynamic potential that is a function of SA, $\Theta$ , p. I encourage the authors to use the more complete: "A Thermodynamic Potential of Seawater in terms of Absolute Salinity, Conservative Temperature, and in situ pressure". I am particularly keen to allow readers who are not oceanographers to find this title offers a better sense for what is proposed.

Many thanks for this comment. We have made the change to the title.

**2.2 Notation introduced in lines 75-80**

In between lines 75-80 there is a single sentence that introduces notation all within the text (i.e., no equation numbers). For those having read other papers in the TEOS-10 family, the notation will be familiar even if one does not like it. Indeed, I have been an author on such papers. Even so, as a reviewer I put on my non-TEOS-10 afficionado hat in hopes of identifying places where readers can be confused.

1. The symbol "t" is almost universally used for time in physics and chemistry. Indeed, you use it for time in the First Law equation (1). However, on lines 77, 78, 79, t is used for in situ temperature measured in Celsius units. Should you wish to retain this non-standard usage, care should be made to reduce confusion when switching between time and temperature, with signposts placed where needed.

Many thanks; this has now been made explicit.

 To is a constant offset from t, presumably since t+To is the absolute thermodynamic temperature with t in Celsius units. That point is never made yet it would serve the reader to make it here. Many thanks; this has now been made explicit.

3. T is a symbol attached as a subscript to the Gibbs potential, and yet it is never defined. Presumably T = t + T0 is the thermodynamic temperature, but the reader should not need to guess. Many thanks; this has now been made explicit.

4. The above incomplete discussion of t and T0 make me wonder if they are even needed at this point of the presentation. Perhaps you prefer to keep them here since you prefer to measure all temperature quantities in

Celsius, and so need the T0 offset. But that is a choice that is not fundamental and can lead to rather awkward equations. Indeed, later in the presentation of the same section, you write  $\nabla(1/T)$  rather than the more awkward  $\nabla(1/(t + T_0))$ .

Many thanks. No changes have been made in regard to this comment. The reason is that potential temperature and Conservative Temperature are both traditionally measured on the Celsius temperature scale. By adopting the temperature notation that we have, our equations are explicit with neither observationalists nor theoreticians being entitled to be confused, even if the notation is at times a bit less neat than it could be. These issues of nomenclature were carefully thought through in the years preceding the publication of the TEOS-10 Manual, and we feel obliged to stick with this internationally accepted definitional document for seawater, humid air and ice Ih.

5. It is stated that subscripts denote partial differentiation, whereas we also find many subscripted symbols that are not derivatives. For example, between lines 75-80 we find Pr, SA, and To. I realize the authors are fond of the subscript shorthand for partial derivatives. Even so, I will poke at them by noting that  $\partial_T$  and  $\partial_{\Theta}$  and  $\partial_P$  require only a bit more ink on the printed page and yet they offer far less room for notational ambiguity.

Many thanks; we have decided to stick with the nomenclature of the TEOS-10 Manual, IOC et al. (2010). .

6. What does "forward expression" mean on line 78? This term is also used later at lines 480, 481, 486, and 523. It is only when reaching lines 486 and 523 that we find out that "forward calculation" means that no iteration is needed. Please define this term at line 78.

Many thanks; this has now been fixed.

7. h is used in two separate equations between lines 75-80, and yet it is not defined until line 90 in a different section. Many thanks; this has now been fixed.

**2.3 Clarifying "inconsistencies"**

On line 80 it is stated that "While the inconsistencies in temperature are small". Although contained in the build up material prior to lines 75, it should be emphasized that the ambiguity is not related to a problem with seawater thermodynamics. Rather, it is that we can compute t and  $\theta$  using either the "forward expression" on line 79 or the implicit Gibbs equation on line 77, and yet these two expressions, in practice, lead to slightly different numerical values. If you clean up lines 75-80, for example by splitting this material into two or three sentences, then the reader will be a bit more clear on the inconsistency.

Many thanks; this suggested rearrangement has been adopted, leading to a clearer presentation.

**2.4 Further citations for First Law**

Line 85-95 presents the First Law of Thermodynamics as applied to moving seawater fluid. Perhaps the most lucid and correct discussion of this equation is given in TEOS-10, with citations given here to the relevant sections. Even so, I recommend including some of the other places that the reader might find it discussed. In particular, it is worth pointing to Chapters 49 and 58 of Landau and Lifshitz (1987) so that our physical chemist reader does not presume TEOS-10 is the first occasion where the First Law was properly derived for a moving multi-component fluid.

Many thanks; these references have now been made.

**2.5 Infer versus deduce**

Line 97 states that Clausius (1876) "deduced" the existence of entropy. I suggest that it is more proper to say that Clausis "inferred" the existence of entropy. It took the later work of Boltzmann to provide a deductive theory for entropy based on mechanistic and probabalistic principles that led to statistical mechanics.

We are not 100% sure whether "inferred" is more accurate than "deduced", but we have changed the word to "inferred".

**2.6 Quasi-static versus reversible**

In much of Section 1.2 (Thermodynamic fundamentals), the authors refer to reversible processes. I instead recommend they use the slightly more general, and useful, term quasi-static, following the usage given in Sections 4.2 and 4.3 of Callen (1985), as well as Sections 2.9 and 2.10 of Reif (1965). In these books, the authors define a quasi-static process as a process that moves along a locus of equilibrium states, and with a quasi-static process approximated by a realizable physical process that occurs through steps that are each arbitrarily close to thermodynamic equilibrium. By formulating the notion of a quasi-static process, we are afforded a precise definition for intensive properties such as in situ temperature, pressure, and chemical potential, whereas such intensive properties are fuzzy concepts for systems out of thermodynamic equilibrium. A reversible process is a quasi-static process that occurs without net entropy change, and yet not all quasi-static processes are reversible since entropy can generally increase in a quasi-static process. Here are the lines of text that prompted me to offer my recommendation to switch from "reversible" to "quasi-static". In practice the distinction might be small. But in the spirit of making the fewest assumptions necessary, I recommend switching from your assumed reversible processes to the more general quasi-static processes.

• Line 102 it is stated that the FTR applies to reversible processes, and yet Callen (1985) emphasizes that the FTR holds for the more general quasi-static processes.

• line 145 states "a seawater parcel is heated reversibly", which I assume means that the parcel's entropy increases

when the surrounding environment entropy decreases by the exact same amount. This sort of process is less general than a quasi-static heating of a parcel, in which we do not need to assume zero net entropy production. 2

• line 149: again change "reversible" to "quasi-static".

• Discussion finishing at line 162. If you replace "reversible" with "quasi-static", then this discussion mirrors that in Section 4.2 and 4.3 of Callen (1985).

• If I am barking up the wrong tree, and you do need reversible, then such disagreement with Callen (1985) and Reif (1965) warrants clarification in your manuscript.

Many thanks for all these suggestions. I think that this issue is perhaps the most confusing issue when thermodynamic theory is used in other sub-fields of physics, such as in our field of fluid dynamics. And what makes it worse is that the issue is hardly ever discussed/confronted. Hence, in the present manuscript we devote considerably more space to discussing exactly which equations must represent reversible processes, and which do not. Writing this down in the manuscript has certainly helped clarify things in the minds of the authors, and we hope that it helps the readers as well. In short, we do not agree with the comments of this reviewer, but we point out that the issue is actually a very complicated one. The issues is plastered over (that is, ignored) in most thermodynamic text books, with one worthy exception being Landau & Lifshitz (1959). In essence, the common equations that are written down for the material derivative of entropy, and also for the production of entropy, are inaccurate equations that have assumed thermodynamic equilibrium in part of their creation, but not in their application to real fluids. Fortunately, we oceanographers have a magic bullet whereby we can avoid this pesky issue. The magic bullet is that we can take advantage of the fact that entropy is a state variable. This means that the evolution of entropy and the production of entropy can be evaluated without encountering these issues. This magic bullet solution is the one adopted by Graham and McDougall (2013) in their evaluation of the non-conservation production of entropy. We hope that the reviewer agrees with our extensive discussion of these issues in the much-expanded section 1.2 of the new manuscript.

**2.7 Gravity with thermodynamics**

When introducing gravity on line 180, you make the distinction between the distribution of in situ temperature and potential temperature in equilibrium. Although a bit tangential to the current discussion, it is worth noting that pressure in thermodynamic equilibrium is not uniform when in a gravity field. Instead pressure is hydrostatically balanced. My motivation for recommending this insertion is that we do not commonly find thermodynamics discussed in the presence of gravity, even though all geophysical fluids are in a gravity field. The notable exeptions include Chapter 9 of Guggenheim (1967), Åò25 of Landau and Lifshitz (1980), and Section 1.8 of Kamenkovich (1977) (I thank Rainer Feistel for pointing these resources to me in email correspondence in 2022).

Many thanks; some references have been added.

**2.8 Criterion of independence**

On line 228 you state that  $h(S_A, T, P)$  and  $\eta(S_A, T, P)$  violate the criterion of independence given that they must satisfy equation (10). You then say that for the Gibbs potential equation (10) is satisfied. I missed how you know that it is not satisfied for  $h(S_A, T, P)$  and  $\eta(S_A, T, P)$ . One or two more sentences might be sufficient.

Many thanks; the word "separately" has been inserted.

Also, equations (9) and (10) have S, P subscripts for the partial derivatives. Should that instead read SA, P? Many thanks; this was indeed a series of a few closely-spaced typos that would have served to confuse and have now

been fixed.

**2.9 Please build a table of symbols!**

There are many symbols that are introduced in the text and in equations. These symbols are easily forgotten, in which case one needs to sift through the text to find them defined. It would greatly help the reader to have a thorough table of symbols for more easy reference.

Many thanks; Table 1 has now been added.

**3 Minor comments**

line 154: I recommend the sentence starting "In practice" should start the beginning of a new paragraph.
lines 176-177 you state "The hat over a variable indicates that it is being expressed as a function of Conservative Temperature (rather than in situ temperature)." This information would be very useful if stated back around line 75-80 where these symbols are first used.

• line 184 you say "here the cup over a variable's name indicates that it is being expressed as a function of entropy". However, in equation (6) you use the same cup symbol to write entropy as a function of SA, h, P. It would be useful to define this notation back at the point of equation (6).

• Equation (8) has a period whereas there should be none.

• Please see the comments from the other reviewer about lines 253 and 256, where S is written when it should be  $S_{A}$ .

• Line 365 there is  $\eta(S_A,\Theta, P)$  and  $\eta(S_A,\Theta, P)$ . I do not know what  $\eta(S_A,\Theta, P)$  is. Perhaps it is a typo, or perhaps in my wished-for table of math symbols I could find it defined.

line 353 starts material that, in my opinion, is better placed in Section 4 when detailing how you compute φ
(SA, Θ, P) in practice.
line 366: The sentence starting with "In summary" is a great start to a new paragraph.

• line 493: the authors make use of a semi-colon here, whereas a colon is more appropriate. I believe there are a few other occasions of this sort, but I did not mark them on my draft. Many thanks; all of the minor comments have now been fixed.

**Referee #3, Remi Tailleux**

Comments on: "A thermodynamic potential of seawater in terms of Conservative Temperature" by McDougall et al.

This is an interesting and useful paper, which I enjoyed reading. I have a few remarks about it though, that the authors may want to address.

1. The paper needs a more scholarly review of the theory of thermodynamic potentials. A very good and lucid reference is Alberty (2001). Such a theory highlights at least three key features that the authors appear to have overlooked. The first one is related to the concept of canonical variables. The second one is the theory of Legendre transforms. The third one is the concept of conjugate thermodynamic variables. The first feature is crucial to mention, because a thermodynamic potential contains all possible thermodynamic function only if expressed in terms of canonical variables. The authors should stress the fact that Conservative Temperature (like potential temperature) is not a canonical variable, which is why two functions (h( $\Theta$ ,S,p) and  $\eta(\Theta$ ,S)) are needed in that case to predict all possible thermodynamic information about the system. The second feature is crucial to mention, because it is the theory of Legendre transform, and the result that a curve can equivalently be described as the envelope of its tangent lines, which serves to establish the equivalence between the different classical thermodynamic potentials. The third feature is crucial to mention to explain why different thermodynamic potentials have different canonical variables. The author may want to mention that introducing Conservative Temperature (or potential temperature) introduces an external environmental parameter into the system (the reference pressure), which augment the phase space from 3 to 4 dimensions.

Thanks for this comment. We have added a substantial amount of new text about these issues in a new section, section 2.1

2. Line 213-214: Note that the FTR follows from this expression for the total derivative of the Gibbs function if and only if one also knows that  $g = h - T\eta$ ; we will return to this later. I don't understand this sentence, because in the theory of thermodynamic potentials, the relation  $g=h-T\eta$  is not a matter of knowledge but of definition, in the sense that the relation defines g as the Legendre transform of h. I don't understand what the authors mean by 'if and only if one also knows that [...]'. Do the authors mean: `if one does not know about the theory of thermodynamic potentials and of Legendre transforms'? That seems odd for a paper about thermodynamic potentials.

Thanks for this comment. Exactly what we mean by the equivalence between the Gibbs function and the new thermodynamic potential is now made much clearer in section 2, starting with a new discussion setting the scene in section 2.1. In summary, what we are addressing is whether the FTR can be deduced from a given thermodynamic potential. In order to do so one needs not only the expression for the total derivative of the thermodynamic potential, but one also needs to know how the potential is defined in terms of internal energy. Without this second piece of information the FTR cannot be confirmed/deduced/derived. This is true even of  $h(S_A, \eta, P)$  and of  $g(S_A, T, P)$ . It is also true of the Helmholtz free energy  $f(S_A, T, v)$  and of the thermodynamic potentials that are newly defined in this paper.

3. I disagree that the new thermodynamic potential is equivalent to the Gibbs function. Indeed, while it is true that the Gibbs function can be recovered from their newly introduced potential via unambiguous mathematical operations (obtain enthalpy and entropy from their new thermodynamic potential; use the result to eliminate conservative temperature and express specific enthalpy in terms of canonical variables; use the Legendre transform to obtain the Gibbs function), it is not possible to recover the newly defined thermodynamic potential from the Gibbs function without introducing external rules along the way. Indeed, since Conservative Temperature is not conjugate to any canonical variable, human intervention is needed to introduce it by specifying the functional relationship linking it to specific entropy and salinity. Actually,  $\hat{\eta}(S_A, \Theta)$  follows from the total derivative of enthalpy in the form (11); see the discussion in the lines following Eqn. (11). This is true of (11) which is written in terms of Conservative Temperature. When the same form of the total differential of enthalpy is re-written for potential temperature,  $\tilde{\eta}(S_A, \theta)$  does not follow from this equation. Rather,  $\tilde{\eta}(S_A, \theta)$  has to be imposed, more in line with what you say here. But this is not true of the Conservative Temperature case. Moreover, as the authors demonstrate, there is no unique way to construct a thermodynamic potential from  $h(S,\Theta,p)$  and  $\eta(S,\Theta)$ , so that human intervention (and ingenuity) is required for that last step. Demonstrating the equivalence between two quantities requires discussion of the steps required to obtain one from the other and conversely.

The form (14) is a guess, a "forward" guess. Then in Appendix D it is shown that the "reverse" derivation of the FTR from (14) also works for this function; that is, there it is shown that knowing the expression for the total differential, and also the definition (14) itself, the FTR follows. This is exactly the same attribute that the  $h(S_A, \eta, P)$ ,  $g(S_A, T, P)$  and  $f(S_A, T, v)$  thermodynamic potentials possess. Ipso facto, this is why we feel justified in calling (14) a thermodynamic potential of seawater.

We note that Legendre transformations define a new thermodynamic function in terms of a new independent variable, and that is exactly what we have achieved in (14), (A1), (A2), (A8) and (A11).

4. While I think that explaining how to construct a thermodynamic potential containing all the thermodynamic information when expressed in terms of a non-canonical variable serves a useful purpose, as it clarifies an issue that others may wonder about, I am sceptical that this is of any practical use. As the authors correctly recognise, all necessary information is contained in  $h(S,\Theta,p)$  and  $\eta(S,\Theta)$ , which are independent of each other. It seems pointless (not to say computationally inefficient) to me to construct a thermodynamic potential just as to be able to say that these two functions can be obtained from it. Indeed, the authors make it clear that the thermodynamic potential that they construct is a purely ad-hoc and arbitrary concept with no particular significance. Therefore, while I agree that it is useful to know that such a thermodynamic potential can be constructed in principle, I disagree that it is useful to construct it in practice. To me, it seems more computationally efficient to construct enthalpy and entropy as function of S, CT, and p, and derive all thermodynamic quantities from these two functions without introducing the thermodynamic potential. For this reason, I think that the authors should reconsider the way that they present their material. I think that the correct

conclusion that derives from their results is: Yes, it is possible to construct a thermodynamic potential in terms of S, CT, and p, and here is how to do it, which is of theoretical interest for the theory of thermodynamic potentials formulated in terms of non-canonical variables, but this does not appear to offer any practical advantage.

We do agree that in the new code, we do not actually use the new thermodynamic potential. Rather we use the two individual polynomials for enthalpy and for entropy. We have now written sections 4 and 5 of the manuscript with more emphasis on this point. It is a good point to make, for otherwise the paper comes across as an oversell of the new material, so thank you for emphasizing this in the review. We knew this but we had not emphasised this aspect sufficiently. This aspect is a really interesting and unique property of Conservative Temperature.

5. Line 535. This provides a theoretical boost to using Conservative Temperature as the temperature variable in physical oceanography as recommended by TEOS-10 (Valladares et al., 2011a,b). I don't understand why for at least two reasons. The first one is that as mentioned before, the authors' thermodynamic potential is clearly an artificial and arbitrary object. As a result, it seems to me that associating Conservative Temperature to an artificial object will just end up highlighting the artificial character of Conservative Temperature, the opposite of a theoretical boost. The second one is that the whole machinery described could obviously be equally applied to potential temperature or any other non-canonical variable. In other words, their results are not specific to Conservative Temperature, so it is unclear why they should favour it more than any other non-canonical variable.

Artificial or not, our new thermodynamic potential, Eqn. (14), works. And importantly, Eqn. (14) is a thermodynamic potential of seawater whose independent temperature variable is Conservative Temperature, a variable that is two orders of magnitude more accurate than is potential temperature when used in ocean models, and in ocean budget studies; see

McDougall, T. J., Barker P. M., Holmes R. M., Pawlowicz R., Griffies, S. M. and Durack, P. J.: The interpretation of temperature and salinity in numerical ocean model output and the calculation of heat fluxes and heat content. *Geoscientific Model Development*, **14**, 6445-6466, https://doi.org/10.5194/gmd-14-6445-2021, 2021.

Also relevant to this discussion is the following response by the authors of the above paper to a reviewer's comments on the above paper,

https://gmd.copernicus.org/preprints/gmd-2020-426/gmd-2020-426-AR1.pdf

Regarding the reviewer's last sentence, we do not intend to imply that the existence of a thermodynamic potential for  $\Theta$  in some way proves that  $\Theta$  is a superior temperature variable; especially so because in this revised manuscript, in the new Eqn. (A8), we have managed to find a respectable thermodynamic potential in terms of potential temperature (I really don't know how Eqn. (A8) escaped my attention for the past nine years; this is embarrassing). Rather the benefits of  $\Theta$  as a temperature variable for use in marine science have been derived, explained and compared in several previous publications. What we are saying in the present manuscript is more like "isn't it nice that a temperature variable  $\Theta$ , which has so many desirable features, now also has attached to it a thermodynamic potential function."